# Predicting nitroimidazole antibiotic resistance mutations in *Mycobacterium tuberculosis* with protein engineering

Brendon M. Lee[1], Liam K. Harold[2], Deepak V. Almeida[3], Livnat Afriat-Jurnou[1,4,5], Htin Lin Aung[2], Brian M. Forde[6], Kiel Hards[2], Sacha J. Pidot[7], F. Hafna Ahmed[1], A. Elaaf Mohamed[1], Matthew C. Taylor[8], Nicholas P. West[6], Timothy P. Stinear[7], Chris Greening[8,9], Scott A. Beatson[6], Eric L. Nuermberger[3], Gregory M. Cook[2], Colin J. Jackson[1] *

1 Research School of Chemistry, Australian National University, Canberra, Australian Capital Territory, Australia, 2 Department of Microbiology and Immunology, University of Otago, Dunedin, New Zealand, 3 Center for Tuberculosis Research, Johns Hopkins University School of Medicine, Baltimore, Maryland, United States of America, 4 MIGAL, Galilee Research Institute, Kiryat Shmona, Israel, 5 Faculty of Sciences and Technology, Tel-Hai Academic College, Upper Galilee, Israel, 6 School of Chemistry and Molecular Biosciences, The University of Queensland, Brisbane, Queensland, Australia, 7 Department of Microbiology and Immunology, University of Melbourne, Melbourne, Victoria, Australia, 8 Land and Water Flagship, The Commonwealth Scientific and Industrial Organisation, Canberra, Australian Capital Territory, Australia, 9 School of Biological Sciences, Monash University, Clayton, Victoria, Australia

* colin.jackson@anu.edu.au

**Data Availability Statement:** All relevant data are within the manuscript and its Supporting Information files.

## Abstract

Our inability to predict which mutations could result in antibiotic resistance has made it difficult to rapidly identify the emergence of resistance, identify pre-existing resistant populations, and manage our use of antibiotics to effectively treat patients and prevent or slow the spread of resistance. Here we investigated the potential for resistance against the new antitubercular nitroimidazole prodrugs pretomanid and delamanid to emerge in *Mycobacterium tuberculosis*, the causative agent of tuberculosis (TB). Deazaflavin-dependent nitroreductase (Ddn) is the only identified enzyme within *M. tuberculosis* that activates these prodrugs, *via* an $F_{420}H_2$-dependent reaction. We show that the native menaquinone-reductase activity of Ddn is essential for emergence from hypoxia, which suggests that for resistance to spread and pose a threat to human health, the native activity of Ddn must be at least partially retained. We tested 75 unique mutations, including all known sequence polymorphisms identified among ~15,000 sequenced *M. tuberculosis* genomes. Several mutations abolished pretomanid and delamanid activation *in vitro*, without causing complete loss of the native activity. We confirmed that a transmissible *M. tuberculosis* isolate from the hypervirulent Beijing family already possesses one such mutation and is resistant to pretomanid, before being exposed to the drug. Notably, delamanid was still effective against this strain, which is consistent with structural analysis that indicates delamanid and pretomanid bind to Ddn differently. We suggest that the mutations identified in this work be monitored for informed use of delamanid and pretomanid treatment and to slow the emergence of resistance.

**Funding:** This work was supported by by NHMRC Project Grant APP1128929 (to C. J. J., C. G., and G. M. C.). In addition, this work was supported by an ARC DECRA Fellowship DE170100310, and NHMRC New Investigator Grant APP1139832 (to C. G.). SAB is supported by an NHMRC Career Development Fellowship GNT1090456 (to SAB). DVA and ELN acknowledge the support of the U.S. National Institutes of Health (R01-AI111992). Websites of funders: https://www.nhmrc.gov.au/ https://www.nhmrc.gov.au/ https://www.nih.gov/ The funders had no role in study design, data collection and analysis, decision to publish, or preparation of the manuscript.

**Competing interests:** I have read the journal's policy and the authors of this manuscript have the following competing interests: ELN reports a research contract from the Global Alliance for TB Drug Development and a research grant from Janssen Pharmaceuticals and collaborates on other research grants with the Global Alliance for TB Drug Development.

## Author summary

Bacterial pathogens often evolve resistance to antibiotics via mutations in the coding sequences of genes–frequently the target, an enzyme that metabolizes or transports the active drug, or an enzyme that activates a prodrug. In the case of tuberculosis, antibiotic resistance is a growing problem, with the rapid emergence of multi-drug resistant strains. New nitroimidazole-based antibiotic prodrugs, such as pretomanid and delamanid have the potential to help infected individuals, but we must guard against the evolution of resistance to these new compounds in *Mycobacterium tuberculosis*. In this report we use protein engineering to identify mutations that could potentially result in antibiotic resistance by knocking out the prodrug activating activity of the deazaflavin dependent nitroreductase (DDN), without completely abolishing its native menaquinone reductase activity. The retention of its native activity is important as DDN appears to be required for *M. tuberculosis* to emerge from hypoxia. Strikingly, when we analysed ~15,000 M. tuberculosis genomes from clinical strains, we identified several that harboured mutations that we identified as abolishing prodrug activation *in vitro*. A hypervirulent Beijing strain N0008 from Vietnam (which had not been exposed to pretomanid in the clinic) was predicted, and confirmed, to be resistant to pretomanid, revealing that resistance to this drug has arisen through genetic drift and not selective pressure in this instance. These data show that by testing potential resistant mutations in the laboratory before large-scale use of antibiotics, we should be able to use them more judiciously in order to slow the spread of resistance.

## Introduction

Tuberculosis (TB) is currently the leading cause of death from a single infectious agent [1]. The limitations of current treatment regimens, combined with the rapid emergence of multi-drug-resistant tuberculosis (MDR-TB) strains, necessitate the development of new drugs. Three new antitubercular agents are now in advanced clinical development: bedaquiline and delamanid [2], which have been conditionally approved for MDR-TB treatment [3,4], and pretomanid [5], which is approved as part of a promising regimen containing bedaquiline and linezolid (http://www.newtbdrugs.org/pipeline/clinical). The nitroimidazoles, delamanid and pretomanid, are prodrugs that are reductively activated in an $F_{420}H_2$-dependent reaction in *Mycobacterium tuberculosis* by deazaflavin-dependent nitroreductase (Ddn) [6,7]. An initial hydride transfer step from the $F_{420}H_2$ cofactor leads to their decomposition into *des*-nitro products and releases reactive nitrogen species that elicit a bactericidal mode-of-action linked to respiratory poisoning and inhibition of mycolic acid synthesis [6–8].

Because pretomanid and delamanid are prodrugs that require activation, mutations that knock-out the activity of Ddn or the biosynthesis or reduction of the enzyme's cofactor ($F_{420}$), could confer resistance. However, the fitness cost of such knockouts may be considerable given that (i) $F_{420}$ has been shown to be conditionally essential to the survival of *M. tuberculosis*, (ii) $F_{420}$ is required by at least 28 different enzymes [9], and (iii) it plays important roles in hypoxic survival, protection against oxidative and nitrosative damage, and evasion of the host immune system [10–12]. Ddn is highly conserved across almost all species of mycobacteria (except *Mycobacterium leprae*), suggesting its physiological role is under strong evolutionary selection [13]. It has been hypothesised that Ddn serves as an $F_{420}H_2$-dependent menaquinone reductase given its membrane localisation and catalytic activity with the synthetic quinone

analogue menadione [12,14], although further work is required to fully define its physiological role.

Despite the recent introduction of nitroimidazoles, cases of acquired clinical resistance, i.e. resistance that occurs during the long treatment of TB infection but is not necessarily transmissible, have already been reported [15,16]. Acquired resistance to pretomanid and delamanid can occur through genetic changes that cause loss of function within the biosynthetic pathway for $F_{420}$ production or in the $F_{420}$-dependent glucose 6-phosphate dehydrogenase (FGD) that catalyzes $F_{420}$ reduction to $F_{420}H_2$ [17,18]. Laboratory studies have also shown that genetic changes that abolish Ddn activity can also confer resistance [18–20]. While such mutations could compromise treatments for individuals that are already infected, transmission of *M. tuberculosis* to healthy individuals after these genetic changes has never been documented. We speculate that in order to for tuberculosis to spread effectively and endanger health, these resistant strains would need to retain sufficient fitness to survive all stages of the lifecycle of *M. tuberculosis*, including recovery from hypoxia. Interestingly, delamanid-resistant isolates with mutations in *ddn* have been recovered from MDR-TB patients who never received delamanid or pretomanid [21,22], raising important questions about the fitness costs associated with such mutations and their potential impact on transmission.

In this study, we analysed Ddn orthologs from related mycobacteria to identify natural sequence variations that make mycobacteria resistant to pretomanid, as well as analysing the genomes of ~15,000 *M. tuberculosis* isolates to identify the spectrum of naturally occurring non-synonymous Ddn polymorphisms. Mutations were then made to Ddn at positions identified in orthologs and through analysis of non-synonymous polymorphisms to analyse their effect on the native activity (quinone reduction) and pretomanid activation. Altogether, 75 mutants, at 47 unique positions within the 151 amino acid Ddn protein were analysed. This identified a number of mutations that prevent pretomanid/delamanid activation, without full loss of the native menaquinone reductase activity. Analysis of complete and partial Ddn knock-outs demonstrated that it is essential for resuscitation from hypoxia, i.e. genetic variants that have lost their nitroimidazole-activation function, but retained their native activity, appear to be sufficiently fit to spread and cause disease in new patients. We examined a hypervirulent strain of *M. tuberculosis* from Vietnam that contains one such mutation (despite never being exposed to pretomanid), which is resistant to pretomanid. Curiously, sensitivity to delamanid activation was not affected in this strain, nor was delamanid activation by the Ddn variant *in vitro*, which is consistent with our structural analysis that suggests it binds to the active site of Ddn in an alternative orientation.

## Results

### The physiological role of Ddn

A role for Ddn as a quinone reductase was previously suggested based on activity with the synthetic menaquinone analogue menadione [12]. Here, we show that purified Ddn catalyzes the $F_{420}H_2$-dependent reduction of menaquinone, which is known to accept electrons from a variety of electron donors and transfer them to terminal oxidases or reductases in mycobacterial respiration [23]. Ddn catalyzed menaquinone-1 reduction *in vitro* with a $k_{cat}/K_M$ value of ~$10^3$ $M^{-1}$ $s^{-1}$ and a $K_M$ of 22.4 ± 3.8 μM (Fig 1a, Table 1). It should be noted that the catalytic rate is likely to be higher in the crowded native environment of the *M. tuberculosis* cell where Ddn and menaquinone are co-localised at the cell membrane [24]. The enzyme kinetics of Ddn are comparable to another mycobacterial menaquinone oxidoreductase, MenJ (Rv0561c), which is found in the menaquinone biosynthetic pathway ($k_{cat}/K_M$ of ~$10^4$ $M^{-1}$ $s^{-1}$ and $K_M$ of 30 μM [25]). The $K_M$ value is significantly lower than the median $K_M$ of enzymes with their native

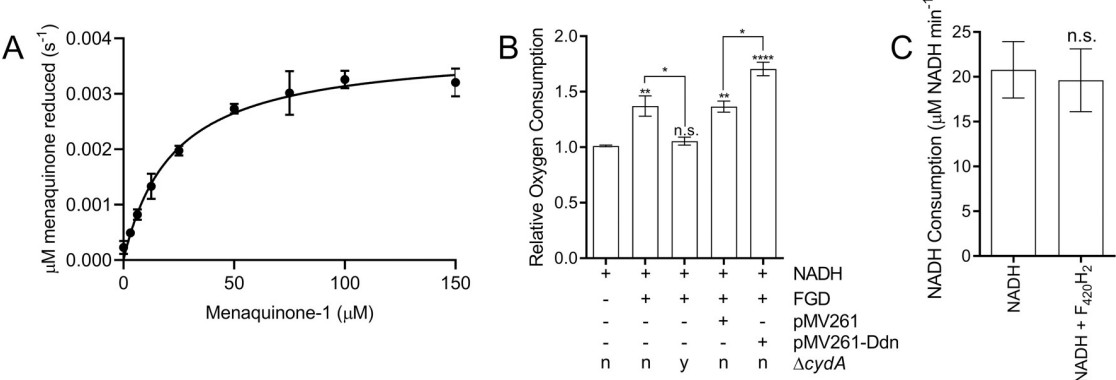

**Fig 1. Ddn is a respiratory primary dehydrogenase that couples $F_{420}H_2$ oxidation to menaquinone reduction. A.** Michaelis-Menten curve of the activity of purified Ddn with menaquinone-1 in the presence of $F_{420}H_2$. Rates were determined by measuring change of absorbance at 420 nm. Error bars indicate SEM (n = 3). **B.** Relative oxygen consumption of *M. smegmatis* membranes. $F_{420}$, NADH, and glucose 6-phosphate were present in all assays, while the availability of the $F_{420}$-dependent glucose 6-phosphate dehydrogenase (FGD) was varied. Membranes were purified from four different genetic backgrounds: wild-type, a transformant with the empty complementation vector pMV261, a transformant with pMV261 expressing the *ddn* gene, and a strain containing a chromosomal cytochrome *bd* oxidase deletion (Δ*cydA*). Error bars indicate SEM (n = 3). Stars above each column indicate a statistically significant difference compared to the first column. Other significant differences are shown by starred lines between columns. * = $p \leq 0.05$, ** = $p \leq 0.01$, **** = $p \leq 0.0001$ n.s = $p > 0.05$ (one-way ANOVA followed by Tukey's multiple comparison test). **C.** Rate of NADH oxidation in the presence and absence of $F_{420}H_2$. Rates were determined by following change of absorbance at 340 nm. n.s indicates $p > 0.05$, two tailed t-test, error bars indicate SEM (n = 3).

substrates (~130 μM), and the $k_{cat}/K_M$ value is within the broad distribution of known enzymes ($10^3$–$10^7$ $M^{-1}$ $s^{-1}$), albeit at the lower end of the range [26,27]. Ddn orthologs encoded by *Mycobacterium smegmatis* (MSMEG_2027, MSMEG_5998) were also able to reduce menaquinone, suggesting that Ddn orthologs have similar physiological roles across the genus (Table 1). Indeed, Ddn and its orthologs are highly conserved and abundant throughout mycobacteria [13].

Having established that Ddn can reduce menaquinone with efficiency that could be physiologically relevant, along with the fact that menaquinone is the only native quinone in *M. tuberculosis* [28–30], we then investigated whether the menaquinone reductase activity of Ddn was coupled to the mycobacterial respiratory chain by comparing the rates of respiratory oxygen consumption of *M. smegmatis* in the presence and absence of a complementation vector expressing *ddn*. We observed that the addition of glucose 6-phosphate (G6P), $F_{420}$, and $F_{420}$ dependent glucose-6-phosphate dehydrogenase (FGD), which catalyzes the reduction of $F_{420}$ to $F_{420}H_2$ for use by $F_{420}H_2$-dependent enzymes such as Ddn [31], resulted in a statistically significant increase in oxygen consumption by mycobacterial membranes when the membranes are activated by NADH (Fig 1b). This was dependent on the presence of cytochrome *bd* oxidase, which utilizes reduced menaquinone (i.e. menaquinol) as an electron source (Fig 1b). The rate of NADH oxidation was not affected by the addition of $F_{420}H_2$ in the absence of *ddn*,

**Table 1. Kinetic parameters of native $F_{420}H_2$-dependent menaquinone reductase activities of Ddn and *M. smegmatis* orthologs in *in vitro* assays.**

| Enzyme | Menaquinone | | |
| --- | --- | --- | --- |
| | $k_{cat}$ ($s^{-1}$) | $K_M$ (μM) | $k_{cat}/K_M$ ($M^{-1}$ $s^{-1}$) |
| Ddn | $1.9 \times 10^{-2} \pm 9.7 \times 10^{-4}$ | $22.4 \pm 3.8$ | $8.6 \times 10^2$ |
| MSMEG_2027 | $5 \times 10^{-3} \pm 1.2 \times 10^{-3}$ | $80.7 \pm 34.4$ | $6.1 \times 10^1$ |
| MSMEG_5998 | $1.2 \times 10^{-2} \pm 1.5 \times 10^{-3}$ | $97.9 \pm 20.4$ | $1.2 \times 10^2$ |

which implies endogenous NADH-dependent oxidases do not contribute to the increased oxygen consumption (Fig 1c). This supports the idea that mycobacteria can couple $F_{420}H_2$ oxidation to $O_2$ reduction through the respiratory chain *via* cytochrome *bd* oxidase and demonstrates that bacteria can use $F_{420}H_2$ as a respiratory electron donor. We also observed that the extent of oxygen consumption was significantly higher in membranes purified from the strain that expressed *ddn*, relative to empty vector controls (Fig 1b), which suggest Ddn serves as a menaquinone reductase in the respiratory chain, with the remaining stimulation attributable to $F_{420}H_2$ oxidation by native Ddn homologs of *M. smegmatis*. Altogether, these results suggest that Ddn likely plays a role in bacterial respiration through reduction of menaquinone.

## Natural sequence variation in Ddn can result in loss of pretomanid activation with retention of the native activity

Having established that Ddn is a menaquinone reductase, we then investigated the effects of natural sequence variation within Ddn orthologs from mycobacteria on the native, and drug-activating, activities. We expressed and purified Ddn orthologs from *M. tuberculosis*, *M. marinum*, *M. smegmatis*, *M. vanbaalenii*, *M. avium* and *M. ulcerans*. All Ddn orthologs catalyzed the reduction of menadione, suggesting that this is a shared physiological function that has been under consistent selective pressure. However, only Ddn from *M. tuberculosis* and *M. marinum* could activate pretomanid (Fig 2a, Table 2). This demonstrates that the native Ddn activity can exist in in the absence of nitroimidazole reductase activity, i.e. the drug-activating activity is not coupled to the native activity. The binding site of Ddn has been defined over several studies, identifying a number of polar amino acids within the substrate binding site that contribute to activity, particularly Y65, S78, Y130, Y133, Y136 (Fig 2b). Three of these binding site residues are fully conserved among the Ddn orthologs tested here, whereas Y65 and Y133 are more variable (Fig 2c). These results show that sequence variation among Ddn orthologs in mycobacteria can result in the loss of the promiscuous, nitroimidazole-activating activity.

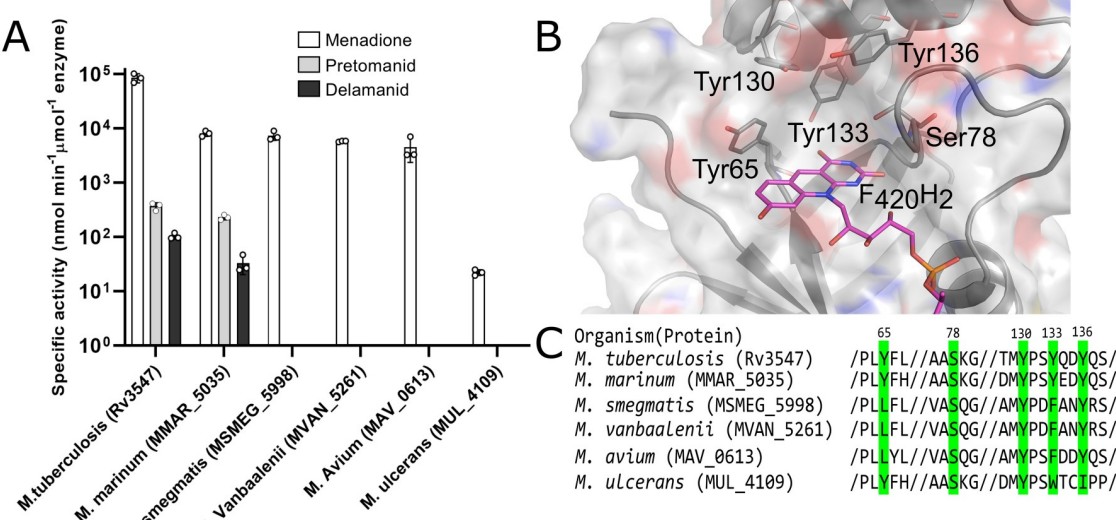

**Fig 2. The activity of Ddn and orthologs from other mycobacterial species and key substrate binding pocket residues. (a)** The activity of Ddn and orthologs with menadione, pretomanid, and delamanid. Error bars show standard deviations from three independent replicates. **(b)** Multiple sequence alignment of Ddn and orthologs. Highlighted residues indicate active site residues and numbers indicate their residue position in Ddn. **(c)** The binding pocket of Ddn consists of $F_{420}/F_{420}H_2$, Tyr65, Ser78, Tyr130 and Tyr136.

**Table 2.  Reduction rates of $F_{420}H_2$-dependent menadione and nitroimidazole reductase activities of Ddn mutants.**

| Ddn Mutants | Menadione | Pretomanid | Delamanid |
|---|---|---|---|
| | Rate of Reaction (nmol min$^{-1}$ μmol$_{enzyme}^{-1}$)[a] | | |
| *M. tuberculosis* mutants selected for pretomanid resistance in laboratory | | | |
| M1T | N/A | N/A | N/A |
| S22L | $5.03 \times 10^2 \pm 4.34 \times 10^1$ | N/D | N/D |
| L49P* | $4.99 \times 10^1 \pm 7.87$ | N/D | N/D |
| L64P | $4.13 \times 10^1 \pm 1.12$ | N/D | N/D |
| W88R* | $2.16 \times 10^1 \pm 7.18 \times 10^{-2}$ | N/D | N/D |
| InsD108 | N/A | N/A | N/A |
| R112W | $1.96 \times 10^3 \pm 2.82 \times 10^2$ | N/D | N/D |
| C149Y | $2.36 \times 10^2 \pm 1.25 \times 10^1$ | N/D | $2.13 \times 10^1 \pm 1.65$ |
| Ddn and orthologs | | | |
| Ddn Mtb | $8.49 \times 10^4 \pm 9.32 \times 10^3$ | $3.66 \times 10^2 \pm 3.3 \times 10^1$ | $1.03 \times 10^2 \pm 7.59$ |
| MMAR_5035 | $7.99 \times 10^3 \pm 9.11 \times 10^2$ | $2.3 \times 10^2 \pm 1.44 \times 10^1$ | $3.26 \times 10^1 \pm 7.11$ |
| MUL_4109 | $2.21 \times 10^1 \pm 2.71$ | N/D | N/D |
| MVAN_5261 | $5.74 \times 10^3 \pm 7.87 \times 10^1$ | N/D | N/D |
| MAV_0613 | $4.51 \times 10^3 \pm 1.24 \times 10^3$ | N/D | N/D |
| MSMEG_5998 | $7.35 \times 10^3 \pm 7.91 \times 10^2$ | N/D | N/D |
| Mutations based on naturally occurring sequence polymorphisms in *M. tuberculosis* | | | |
| P6S | $3.74 \times 10^4 \pm 8.74 \times 10^3$ | $3.29 \times 10^2 \pm 1.63 \times 10^1$ | $8.94 \times 10^1 \pm 4.64$ |
| P6T | $5.07 \times 10^4 \pm 5.16 \times 10^3$ | $5.07 \times 10^2 \pm 3.18 \times 10^1$ | $1.13 \times 10^2 \pm 4.17$ |
| P6L | $8.24 \times 10^4 \pm 1.46 \times 10^3$ | $1.94 \times 10^2 \pm 2.84$ | $5.84 \times 10^1 \pm 1.23 \times 10^1$ |
| M21T | $1.52 \times 10^4 \pm 1.26 \times 10^3$ | $3.65 \times 10^2 \pm 8.99$ | $7.29 \times 10^1 \pm 3.60$ |
| R23L | $4.59 \times 10^4 \pm 3.28 \times 10^3$ | $4.95 \times 10^2 \pm 1.02 \times 10^1$ | $1.20 \times 10^2 \pm 3.93$ |
| R23W | $9.86 \times 10^4 \pm 9.98 \times 10^3$ | $2.00 \times 10^2 \pm 1.37 \times 10^1$ | $9.15 \times 10^1 \pm 3.25 \times 10^1$ |
| T26P | $3.29 \times 10^3 \pm 3.43 \times 10^2$ | $6.21 \times 10^1 \pm 9.18$ | $5.82 \pm 3.10$ |
| W27C | $1.44 \times 10^4 \pm 2.10 \times 10^2$ | $1.66 \times 10^2 \pm 5.98$ | $4.19 \times 10^1 \pm 1.91$ |
| Y29H | $1.51 \times 10^4 \pm 1.54 \times 10^3$ | $5.11 \times 10^1 \pm 7.60$ | $1.77 \times 10^1 \pm 4.81$ |
| Y29S | $7.25 \times 10^3 \pm 6.67 \times 10^2$ | $3.88 \times 10^1 \pm 9.51 \times 10^{-1}$ | $7.15 \pm 2.69$ |
| R30S | $6.04 \times 10^4 \pm 1.86 \times 10^3$ | $1.44 \times 10^2 \pm 1.81 \times 10^1$ | $7.11 \times 10^1 \pm 7.87$ |
| G34E | $6.65 \times 10^4 \pm 3.10 \times 10^3$ | $7.39 \times 10^1 \pm 4.72$ | $5.29 \times 10^1 \pm 6.98$ |
| G34R | $9.14 \times 10^3 \pm 7.79 \times 10^2$ | $3.36 \times 10^1 \pm 5.92$ | $4.19 \times 10^1 \pm 2.77$ |
| G36V | $4.78 \times 10^4 \pm 5.09 \times 10^2$ | $8.16 \times 10^1 \pm 1.36 \times 10^1$ | $4.91 \times 10^1 \pm 1.88$ |
| P45L | $7.69 \times 10^2 \pm 5.92 \times 10^1$ | $8.29 \times 10^1 \pm 1.52 \times 10^1$ | $7.78 \times 10^1 \pm 1.05$ |
| L49P | $4.99 \times 10^1 \pm 7.87$ | N/D | N/D |
| T50P | $3.36 \times 10^2 \pm 6.07 \times 10^1$ | $5.37 \times 10^1 \pm 7.75$ | $5.65 \times 10^1 \pm 5.90$ |
| T51P | $1.35 \times 10^3 \pm 1 \times 10^2$ | $3.70 \times 10^1 \pm 4.80$ | $2.87 \times 10^1 \pm 5.04 \times 10^{-1}$ |
| T52N | $7.61 \times 10^4 \pm 1.01 \times 10^4$ | $2.74 \times 10^2 \pm 6.21$ | $1.19 \times 10^2 \pm 7.13$ |
| T52P | $9.66 \times 10^3 \pm 3.38 \times 10^2$ | $8.73 \times 10^1 \pm 6.03$ | $5.07 \times 10^1 \pm 3.04$ |
| T56P | $1.20 \times 10^4 \pm 2.84 \times 10^3$ | $1.01 \times 10^1 \pm 8.60 \times 10^{-1}$ | $2.79 \times 10^1 \pm 9.43 \times 10^{-1}$ |
| G57A | $5.60 \times 10^4 \pm 1.47 \times 10^3$ | $2.65 \times 10^2 \pm 1.32 \times 10^1$ | $9.69 \times 10^1 \pm 9.43$ |
| V61G | $4.98 \times 10^4 \pm 3.05 \times 10^3$ | $1.01 \times 10^2 \pm 2.25$ | $5.46 \times 10^1 \pm 6.00$ |
| N62D | $1.22 \times 10^3 \pm 1.62 \times 10^2$ | $1.68 \times 10^1 \pm 9.60 \times 10^{-1}$ | $2.19 \times 10^1 \pm 1.80$ |
| Y65S | $1.12 \times 10^3 \pm 8.63 \times 10^1$ | N/D | N/D |
| L67P | $6.86 \times 10^2 \pm 8.52$ | $3.58 \times 10^1 \pm 6.73 \times 10^{-1}$ | $1.55 \times 10^1 \pm 1$ |
| D69N | $6.33 \times 10^4 \pm 1.28 \times 10^3$ | $2.94 \times 10^2 \pm 4.06$ | $1.25 \times 10^2 \pm 2.00 \times 10^1$ |
| G71R | $9.49 \times 10^4 \pm 4.63 \times 10^3$ | $3.71 \times 10^2 \pm 6.81$ | $1.31 \times 10^2 \pm 1.86$ |
| R72Q | $5.20 \times 10^4 \pm 9.28 \times 10^2$ | $2.11 \times 10^2 \pm 1.14 \times 10^1$ | $8.32 \times 10^1 \pm 4.33$ |

(*Continued*)

**Table 2.** (*Continued*)

| Ddn Mutants | Rate of Reaction (nmol min$^{-1}$ µmol$_{enzyme}$$^{-1}$)$^{a}$ | | |
|---|---|---|---|
| | Menadione | Pretomanid | Delamanid |
| R72W | $5.08 \times 10^4 \pm 1.31 \times 10^3$ | $2.27 \times 10^2 \pm 1.47$ | $1.07 \times 10^2 \pm 1.99$ |
| S78Y | $1.41 \times 10^1 \pm 3.05$ | N/D | $1.21 \times 10^1 \pm 2.72$ |
| K79Q | $7.95 \times 10^3 \pm 8.05 \times 10^2$ | N/D | N/D |
| G81S | $1.52 \times 10^4 \pm 1.26 \times 10^3$ | $3.00 \times 10^1 \pm 5.93$ | $2.28 \times 10^1 \pm 2.81$ |
| E83D | $6.93 \times 10^4 \pm 3.28 \times 10^3$ | $4.22 \times 10^2 \pm 6.09 \times 10^1$ | $1.56 \times 10^2 \pm 1.04 \times 10^1$ |
| W88R | $2.16 \times 10^1 \pm 7.18 \times 10^{-2}$ | N/D | N/D |
| L90V | $6.61 \times 10^4 \pm 2.60 \times 10^3$ | $2.49 \times 10^2 \pm 1.23 \times 10^1$ | $7.68 \times 10^1 \pm 8.40$ |
| N91T | $2.24 \times 10^3 \pm 5.07 \times 10^2$ | $3.19 \times 10^1 \pm 5.85$ | $3.00 \times 10^1 \pm 1.45$ |
| I102V | $7.60 \times 10^4 \pm 4.82 \times 10^3$ | $2.88 \times 10^2 \pm 1.10 \times 10^1$ | $8.25 \times 10^1 \pm 1.54 \times 10^1$ |
| E105Q | $1.08 \times 10^5 \pm 1.60 \times 10^4$ | $3.49 \times 10^2 \pm 7.56$ | $1.23 \times 10^2 \pm 1.78$ |
| A111V | $5.62 \times 10^2 \pm 2.25 \times 10^1$ | $4.22 \times 10^1 \pm 7.77$ | $2.24 \times 10^1 \pm 3.64$ |
| D113N | $3.45 \times 10^4 \pm 1.31 \times 10^3$ | $1.94 \times 10^2 \pm 1.26 \times 10^1$ | $4.48 \times 10^1 \pm 5.85$ |
| E117K | $7.07 \times 10^4 \pm 2.11 \times 10^3$ | $3.13 \times 10^2 \pm 5.83$ | $9.15 \times 10^1 \pm 8.51$ |
| P124S | $7.18 \times 10^4 \pm 2.48 \times 10^3$ | $2.92 \times 10^2 \pm 3.35$ | $8.75 \times 10^1 \pm 1.03 \times 10^1$ |
| Y133C | $1.82 \times 10^3 \pm 1.54 \times 10^2$ | N/D | $3.90 \pm 1.95$ |
| T140I | $7.28 \times 10^4 \pm 4.09 \times 10^2$ | $1.66 \times 10^2 \pm 2.40 \times 10^1$ | $1.03 \times 10^2 \pm 3.50$ |
| V147M | $4.38 \times 10^4 \pm 4.29 \times 10^3$ | $1.51 \times 10^2 \pm 5.97$ | $6.40 \times 10^1 \pm 7.73$ |
| Potential spontaneous mutations within the substrate binding site of Ddn | | | |
| Y65L | $4.07 \times 10^4 \pm 3.22 \times 10^3$ | N/D | $6.66 \times 10^1 \pm 5.23$ |
| Y65M | $6.72 \times 10^4 \pm 5.78 \times 10^3$ | N/D | $2.25 \times 10^1 \pm 2.82$ |
| Y65S* | $1.12 \times 10^3 \pm 8.63 \times 10^1$ | N/D | N/D |
| Y65C | $3.11 \times 10^4 \pm 9.27 \times 10^2$ | N/D | $4.43 \times 10^1 \pm 6.39$ |
| Y65F | $3.59 \times 10^4 \pm 2.09 \times 10^3$ | $2.93 \times 10^2 \pm 1.82 \times 10^1$ | $2.41 \times 10^2 \pm 1.35 \times 10^1$ |
| S78A | $4.17 \times 10^4 \pm 2.47 \times 10^3$ | N/D | $1.20 \times 10^1 \pm 1.68$ |
| S78C | $4.08 \times 10^3 \pm 1.54 \times 10^2$ | N/D | $2.40 \times 10^1 \pm 4.13$ |
| S78T | $6.15 \times 10^4 \pm 6.77 \times 10^3$ | N/D | $1.90 \times 10^1 \pm 1.75$ |
| S78V | $1.12 \times 10^3 \pm 2.00 \times 10^2$ | N/D | $6.54 \pm 3.87$ |
| S78Y* | $1.41 \times 10^1 \pm 3.05$ | N/D | $1.21 \times 10^1 \pm 2.72$ |
| Y130C | $3.31 \times 10^3 \pm 3.92 \times 10^2$ | $7.62 \times 10^1 \pm 2.49$ | $5.15 \times 10^1 \pm 3.85$ |
| Y130D | $7.08 \times 10^3 \pm 2.17 \times 10^3$ | $3.48 \times 10^1 \pm 2.57$ | $2.42 \times 10^1 \pm 1.3$ |
| Y130F | $1.41 \times 10^4 \pm 2.12 \times 10^3$ | $2.75 \times 10^1 \pm 7.45$ | $4.92 \times 10^1 \pm 4.82 \times 10^{-1}$ |
| Y130H | $4.63 \times 10^3 \pm 4.05 \times 10^2$ | $4.02 \times 10^1 \pm 8.18$ | $3.19 \times 10^1 \pm 1.02 \times 10^1$ |
| Y130N | $3.65 \times 10^3 \pm 2.07 \times 10^2$ | $7.66 \times 10^1 \pm 8.82$ | $3.57 \times 10^1 \pm 4.03$ |
| Y130S | $1.35 \times 10^4 \pm 1.46 \times 10^3$ | $2.19 \times 10^2 \pm 2.24 \times 10^1$ | $3.66 \times 10^1 \pm 1.31 \times 10^1$ |
| Y130W | $1.28 \times 10^4 \pm 1.39 \times 10^3$ | $9.77 \times 10^1 \pm 2.57$ | $5.53 \times 10^1 \pm 4.63$ |
| Y133C* | $1.82 \times 10^3 \pm 1.54 \times 10^2$ | N/D | $3.90 \pm 1.95$ |
| Y133F | $4.08 \times 10^4 \pm 3.03 \times 10^3$ | $1.41 \times 10^2 \pm 6.54$ | $7.82 \times 10^1 \pm 3.58$ |
| Y133L | $8.87 \times 10^4 \pm 1.10 \times 10^3$ | N/D | $2.78 \times 10^1 \pm 5.64$ |
| Y133M | $5.82 \times 10^4 \pm 3.57 \times 10^3$ | N/D | $1.64 \times 10^1 \pm 2.32$ |
| Y133W | $1.45 \times 10^4 \pm 6.57 \times 10^2$ | N/D | $7.80 \pm 4.47 \times 10^{-1}$ |
| Y136E | $1.24 \times 10^2 \pm 6.10$ | N/D | $4.51 \times 10^1 \pm 1.93$ |
| Y136T | $6.86 \times 10^2 \pm 1.85 \times 10^1$ | N/D | $3.67 \pm 1.17 \times 10^{-1}$ |
| Y136F | $5.08 \times 10^4 \pm 5.64 \times 10^3$ | $3.88 \times 10^2 \pm 3.07 \times 10^1$ | $1.05 \times 10^2 \pm 1.29 \times 10^1$ |

(*Continued*)

**Table 2.** (Continued)

| | Rate of Reaction (nmol min$^{-1}$ $\mu$mol$_{enzyme}^{-1}$)[a] | | |
|---|---|---|---|
| Ddn Mutants | Menadione | Pretomanid | Delamanid |
| Y136S | $5.13 \times 10^2 \pm 1.38 \times 10^1$ | N/D | N/D |

N/D–Below limit of detection, N/A–Enzyme was unable to express,

*—found in naturally occurring sequence polymorphisms

[a]–Data from this table is also used in Table 3, and Figs 4 and 6

Having demonstrated that mutations can cause the loss of nitroimidazole-reducing activity *in vitro*, we investigated whether this corresponded to differences in nitroimidazole susceptibility *in vivo*. *M. tuberculosis* (H37Rv) and *M. marinum*, which were the only two species that encoded Ddn orthologs with in vitro pretomanid activation activity, were found to be susceptible to pretomanid and delamanid treatment (Table 3). In contrast, species in which the Ddn orthologs did not exhibit pretomanid activation activity (*M. smegmatis*, *M. ulcerans*, *M. avium*) have been shown to be naturally resistant to pretomanid [5,32,33]. This analysis confims that the prodrug-activating activity of Ddn from *M. tuberculosis* H37Rv and *M. marinum* must result from sequence differences to the other Ddn orthologs tested.

Genome sequences of *M. tuberculosis* were then searched to identify nonsynonymous sequence polymorphisms within the *ddn* gene. We found that around 1.5% (219/14,876) of presumptive *M. tuberculosis* genomes screened encoded a non-synonymous mutation in *ddn*, including several that have arisen independently in unrelated strains (S1 Table). Altogether, we identified 46 non-synonymous substitutions and 2 deletions in *ddn*, distributed throughout the *M. tuberculosis* phylogeny (Table 4).

All 46 mutants found in our genomic survey were expressed, purified, and assayed with menadione, pretomanid, and delamanid (Table 2, Figs 3 and 4). Every mutant tested was able to reduce menadione, which is consistent with selective pressure for maintenance of the physiological function of Ddn. The majority of the mutants could also reduce/activate pretomanid and delamanid. This is not unexpected, as these *M. tuberculosis* strains have not (to our knowledge) been exposed to either drug and have therefore not been under selective pressure to develop resistance. However, of the 46 mutants, several mutants did not activate pretomanid (L49P, S78Y, K79Q, W88R, and Y133C) indicating that any strain of *M. tuberculosis* with these mutations would be unable to activate the drug. Indeed, an *M. tuberculosis* strain haboring the

**Table 3. MICs of pretomanid and delamanid against different mycobacterial strains.**

| | Pretomanid | | Delamanid | |
|---|---|---|---|---|
| Strain | Reduction (nmol min$^{-1}$ $\mu$mol$_{enzyme}^{-1}$) | MIC ($\mu$g mL$^{-1}$) | Reduction (nmol min$^{-1}$ $\mu$mol$_{enzyme}^{-1}$) | MIC ($\mu$g mL$^{-1}$) |
| *M. tuberculosis* H37Rv[a] | 366 ± 33[c] | 4 | 103 ± 7.59 [c] | 32 |
| *M. tuberculosis* N0008[a,b] | N/D | 256 | 12.1 ± 2.72 [c] | 32 |
| *M. marinum* M | 230 ± 14.4 [c] | 16 | 326 ± 7.11 [c] | 16 |
| *M. smegmatis* mc$^2$155 | N/D | >50* | N/D | >50* |

N/D–Below limit of detection

[a]MIC was determined by reazurin assay, measuring the minimal cell death concentration

[b]S78Y mutation

[c]Data used previously in Table 2, and Figs 4 and 6

*MICs obtained from Upton, A. et al. [33]

**Table 4. Distribution of Ddn alleles in the major *M. tuberculosis* lineages.** Distribution of Ddn alleles in five major lineages of *M. tuberculosis*.

| Ddn allele | #Strains | Lineage(s) |
|---|---|---|
| A111V | 12 | Lineage 4 (Euro American) |
| D113N | 7 | Lineage 5 (West Africa) |
| D69N | 1 | Lineage 4 (Euro American) |
| E105Q | 1 | Lineage 4 (Euro American) |
| E117K | 1 | Lineage 2 (East Asian-Beijing) |
| E83D | 4 | Lineage 4 (Euro American) |
| G34R | 11 | Lineage 4 (Euro American) |
| G34E | 2 | Lineage 4 (Euro American) |
| G36V | 1 | Lineage 1 (Indo-Oceanic) |
| G57A | 5 | Lineage 1(Indo-Oceanic),Animal Lineage |
| G71R | 1 | Lineage 4 (Euro American) |
| G81S | 9 | Lineage 2 (East Asian-Beijing) |
| I102V | 1 | Lineage 4 (Euro American) |
| K79Q | 1 | Lineage 4 (Euro American) |
| L49P | 16 | Lineage 2 (East Asian-Beijing) |
| L67P | 2 | Lineage 4 (Euro American) |
| L90V | 1 | Animal Lineage |
| M21T | 1 | Lineage 3(East African-Indian) |
| N62D | 1 | Lineage 4 (Euro American) |
| N91T | 1 | Lineage 4 (Euro American) |
| P124S | 1 | Lineage 2 (East Asian-Beijing) |
| P45L | 2 | Lineage 4 (Euro American),Lineage 1 (Indo-Oceanic) |
| P6L | 2 | Lineage 2 (East Asian-Beijing) |
| P6S | 9 | Lineage 1 (Indo-Oceanic) |
| P6T | 2 | Lineage 3(East African-Indian) |
| P86_M87del | 22 | Lineage 4 (Euro American) |
| R23L | 1 | Lineage 4 (Euro American) |
| R23W | 14 | Lineage 4 (Euro American) |
| R30S | 3 | Lineage 2 (East Asian-Beijing) |
| R72Q | 3 | Lineage 4 (Euro American) |
| R72W | 34 | Lineage 1 (Indo-Oceanic) |
| S78Y | 3 | Lineage 2 (East Asian-Beijing) |
| T140I | 1 | Lineage 4 (Euro American) |
| T26P | 2 | Lineage 4 (Euro American) |
| T50P | 2 | Lineage 2 (East Asian-Beijing) |
| T51P | 1 | Lineage 2 (East Asian-Beijing) |
| T52N | 1 | Lineage 2 (East Asian-Beijing) |
| T52P | 2 | Lineage 3 (East African-Indian),Lineage 4 (Euro American) |
| T56P | 5 | Lineage 4 (Euro American),Lineage 3 (East African-Indian) |
| TTT50PPP | 1 | Lineage 3 (East African-Indian) |
| V147M | 1 | Lineage 1 (Indo-Oceanic) |
| V61G | 5 | Lineage 4 (Euro American) |
| W27C | 1 | Lineage 2 (East Asian-Beijing) |
| W88R | 2 | Lineage 4 (Euro American) |
| Y122_M129del | 5 | Lineage 4 (Euro American) |
| Y133C | 1 | Lineage 1 (Indo-Oceanic) |

*(Continued)*

**Table 4.** (Continued)

| Ddn allele | #Strains | Lineage(s) |
|---|---|---|
| Y29del | 2 | Lineage 2 (East Asian-Beijing) |
| Y29H | 1 | Lineage 4 (Euro American) |
| Y29S | 2 | Lineage 4 (Euro American) |
| Y65S | 3 | Lineage 4 (Euro American) |
| TG56PA | 6 | Lineage 3 (East African-Indian),Lineage 4 (Euro American) |

W88R mutation has been shown to be resistant to pretomanid *in vitro* [20], and the L49P mutant was obtained from an *in vivo* resistance selection experiment [34]. It is notable that the S78Y and Y133C mutants retained the ability to activate delamanid.

The S78Y polymorphism is found in the genome of N0008 [35], a clinical isolate of the hypervirulent Beijing family [15], and in two other genomes (SRA accessions:ERR718320 and ERR751847) that are phylogenetically closely related, indicating a shared evolutionary history and suggesting that there was no substantial selective pressure to eliminate this mutation, i.e. its fitness cost must have been relatively small (Fig 5). We obtained the Beijing strain N0008 to investigate whether the S78Y mutation in the *ddn* gene, which results in loss of pretomanid reduction *in vitro* (Table 3), corresponded to resistance to pretomanid *in vivo*. No other genetic changes previously observed to cause nitroimidazole resistance (e.g. loss of $F_{420}$ biosynthetic genes) were apparent in the genome of N0008. We observed a 64-fold increase in minimum inhibitory concentration (MIC) of pretomanid against N0008 (256 μg mL$^{-1}$) compared to H37Rv (4 μg mL$^{-1}$). This suggests SNPs, such as L49P, S78Y and W88R, in the *ddn* gene can confer resistance to pretomanid and confirms that transmissible pretomanid-resistant populations of *M. tuberculosis* already exist.

## The potential for spontaneous pretomanid resistance mutations to arise in *M. tuberculosis*

Previous studies have used laboratory evolution or engineering to investigate the potential for pathogens to evolve resistance to antibiotics [36,37]. We took a similar approach here, using structure-guided mutagenesis to investigate the robustness of the nitroreductase activity of Ddn to mutations. The binding site of Ddn has been defined over several studies, identifying a number of polar amino acids within the substrate binding site that contribute to activity, particularly Y65, S78, Y130, Y133, Y136. Three of these binding site residues are fully conserved among the Ddn orthologs tested here, whereas Y65 and Y133 are more variable (Fig 2c). We made, expressed and purified 26 mutants (including the Y65S, S78Y, and Y133C sequence differences observed some of the other Ddn orthologs) of these five key substrate binding residues (Y65, S78, Y130, Y133, Y136) (Fig 3), to test how spontaneous mutations at these positions affect nitroimidazole activation. Across the 26 mutants tested, all retained significant levels of the native activity, while 16 did not display detectable pretomanid activation (Table 2, Fig 6). In other words, while the native activity was not greatly affected by sequence variation, the promiscuous nitroreductase activity was extremely sensitive to mutation.

To estimate the rate at which mutations could spontaneously arise at one of the positions identified in this work, we used our genomic survey of *ddn* in *M. tuberculosis* to estimate the level of sequence variation in *ddn*. This revealed that virtually every distinct lineage of *M. tuberculosis* contains SNPs in *ddn*; given the burden of TB in the world and the potential for widespread pretomanid administration, alongside the apparent sensitivity of the promiscuous pretomanid activation activity to such SNPs in contrast to the native activity, it appears that

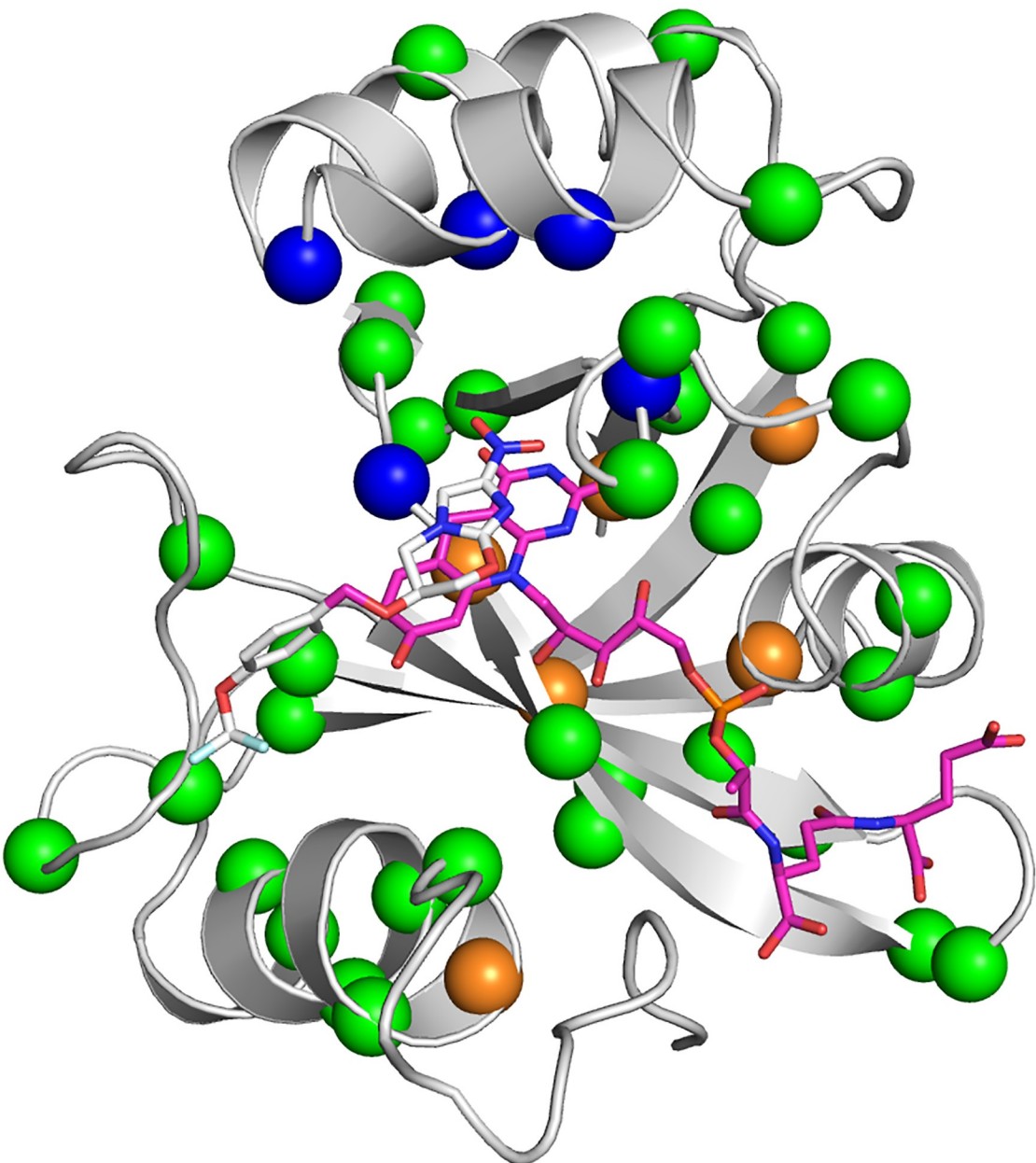

**Fig 3. Mutations to Ddn that were made in this study.** The holo-enzyme structure of Ddn with a reconstructed N-terminus and $F_{420}$ and pretomanid bound is shown, with mutations highlighted from mutants selected for pretomanid resistance in laboratory (orange spheres), non-synonymous mutations found in sequenced *M. tuberculosis* strains (green spheres), and potential spontaneous mutations within the substrate binding site of Ddn (blue spheres).

spontaneous resistance enabling mutations of *ddn* could readily occur and spread with sufficient selective pressure.

## The molecular basis of resistance

The effects of the mutations shown in Table 2 are generally consistent with our mechanistic understanding of Ddn from mutagenesis and computational simulation [7,38]. For example,

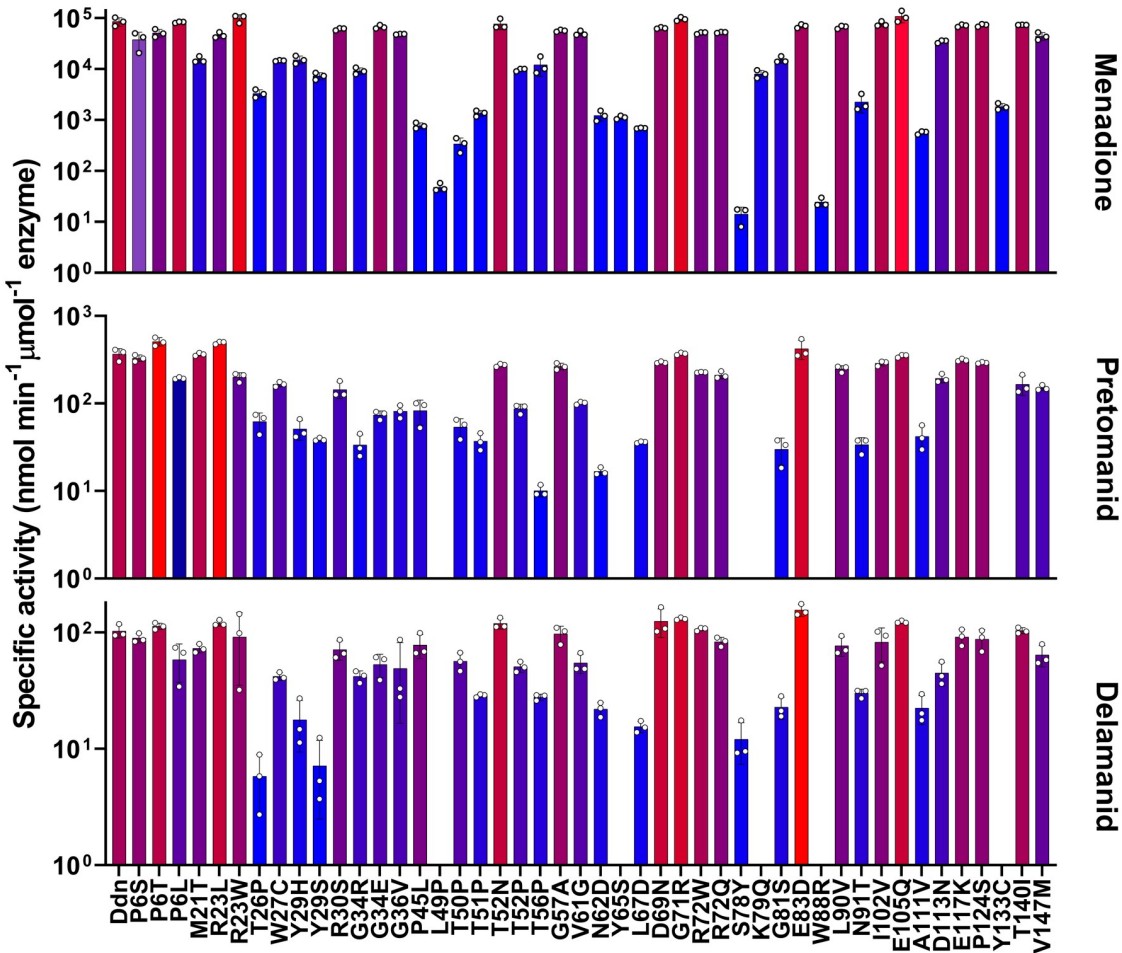

**Fig 4. Natural sequence polymorphisms found in *M. tuberculosis* stains.** The activity of Ddn mutants with menadione, pretomanid, and delamanid. Bar color represents level of activity related to the rest of the graph with the highest activity colored red and the lowest activity colored blue. Error bars show standard deviations from three independent replicates. Data from this Figure is also present in Tables 2 and 3.

S78 is thought to interact with the nitro-moiety of pretomanid and to stabilize the transition state; none of the S78 mutants retained pretomanid nitroreductase activity, suggesting this interaction is particularly important. Y65, Y130, Y133 and Y136 are known to form a hydrophobic wall in the binding site, which can move during the catalytic cycle to shield the active site from solvent and thereby facilitate pretomanid reduction [38]. This is in keeping with the observation that tyrosine to phenylalanine mutations at positions 65, 133 and 136 were essentially neutral, whereas substitution with other residues (Met, Leu, Cys, Trp, Thr, Ser, Glu) led to loss of pretomanid activation. Menadione reduction by Ddn is less susceptible to loss of activity through mutation, which is consistent with work showing native activities are substantially more robust to mutation than promiscuous activities (such as pretomanid activation) [39], as well as the observation that menadione is more chemically labile.

## Delamanid activation is less susceptible to resistance mutations

Of the 75 mutants we made and tested, 25 did not reduce pretomanid at detectable levels but only 10 lost the ability to reduce delamanid. Thus, although pretomanid and delamanid are

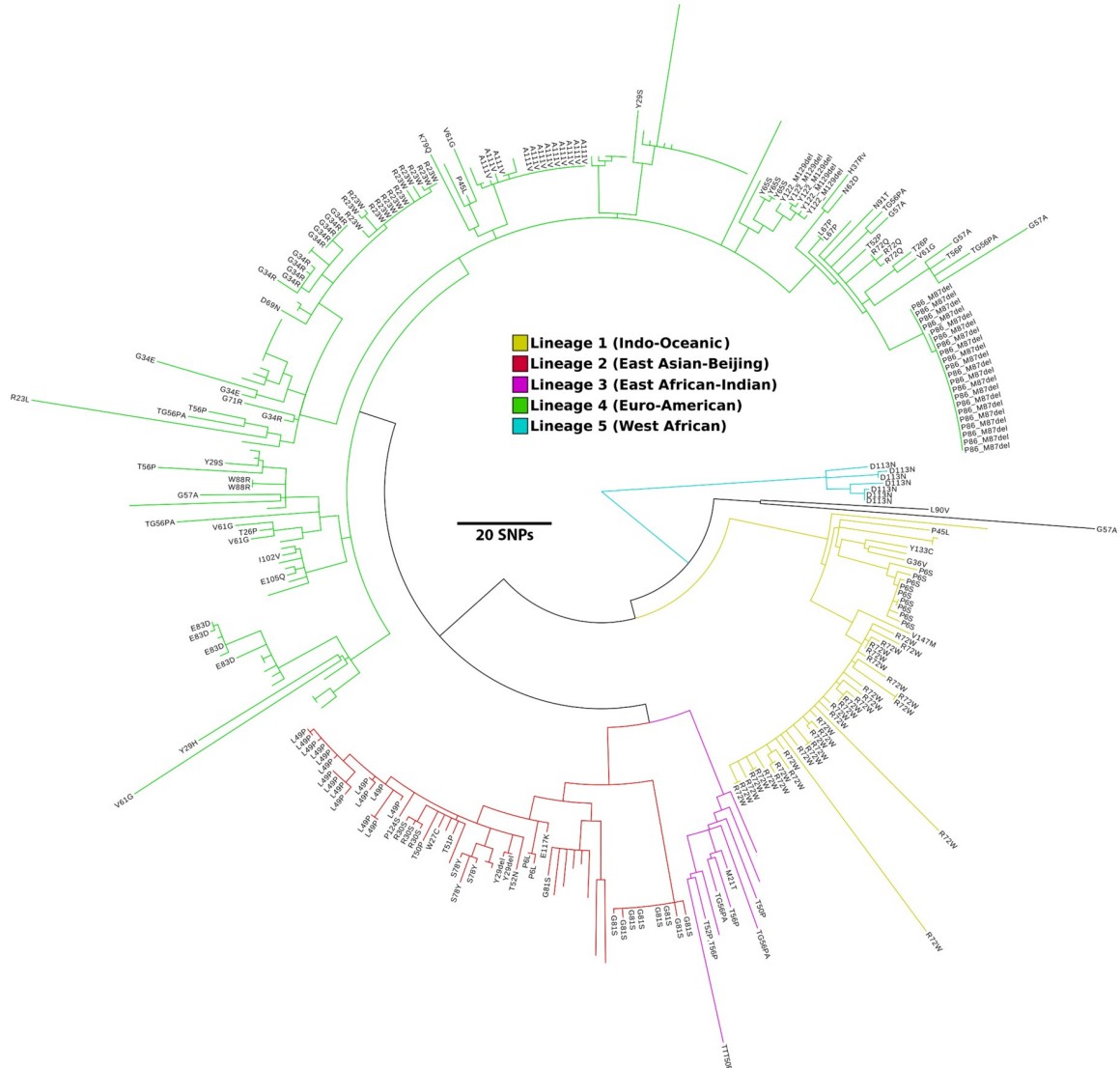

**Fig 5. Phylogeny of *M. tuberculosis* strains with sequence polymorphisms in Ddn.** Maximum-likelihood phylogenetic tree of 332 *M. tuberculosis* strains built using 2,099 non-recombinant core genome SNPs (relative to H37Rv). The tree shows all *M. tuberculosis* strains for which public genome data is available in Genbank or the short read archive (SRA) that have either synonymous and non-synonymous mutations in Ddn (relative to H37Rv). The amino acid changes of the non-synonymous mutations are indicated on the branch tips. Blank tips represent strains with synonymous mutations in Ddn. Branches are coloured by Lineage: Lineage 1 (Indo-Oceanic), yellow; Lineage 2 (East Asian-Beijing), red; Lineage 3 (East African-Indian), purple; Lineage 4 (Euro-American), green; Lineage 5 (West African), teal. The scale bar indicates branch length in number of SNPs. Genome alignments, recombination filtering and phylogenetic reconstruction were done using Parsnp, Gubbins and RaxML, respectively. The phylogenetic tree was visualised using Figtree version 1.4.3 (https://tree.bio.ed.ac.uk/software/figtree).

superficially similar, they must interact with Ddn differently. These enzymatic data extend to whole cell activity, as we observed that the N0008 strain harbouring the S78Y mutation was resistant to pretomanid but remained susceptible to delamanid (Table 3). To better understand the molecular basis for the differential effects of mutations on pretomanid and delamanid activation, we used molecular docking (we were unable to capture ternary crystal structures with ligand and cofactor) to obtain low-energy poses of these substrates in the crystal structure of Ddn. The binding of delamanid is predicted to be different than that of pretomanid (Fig 7B and 7C), with the dual methyl and phenoxy-methyl substituents on the oxazole ring preventing

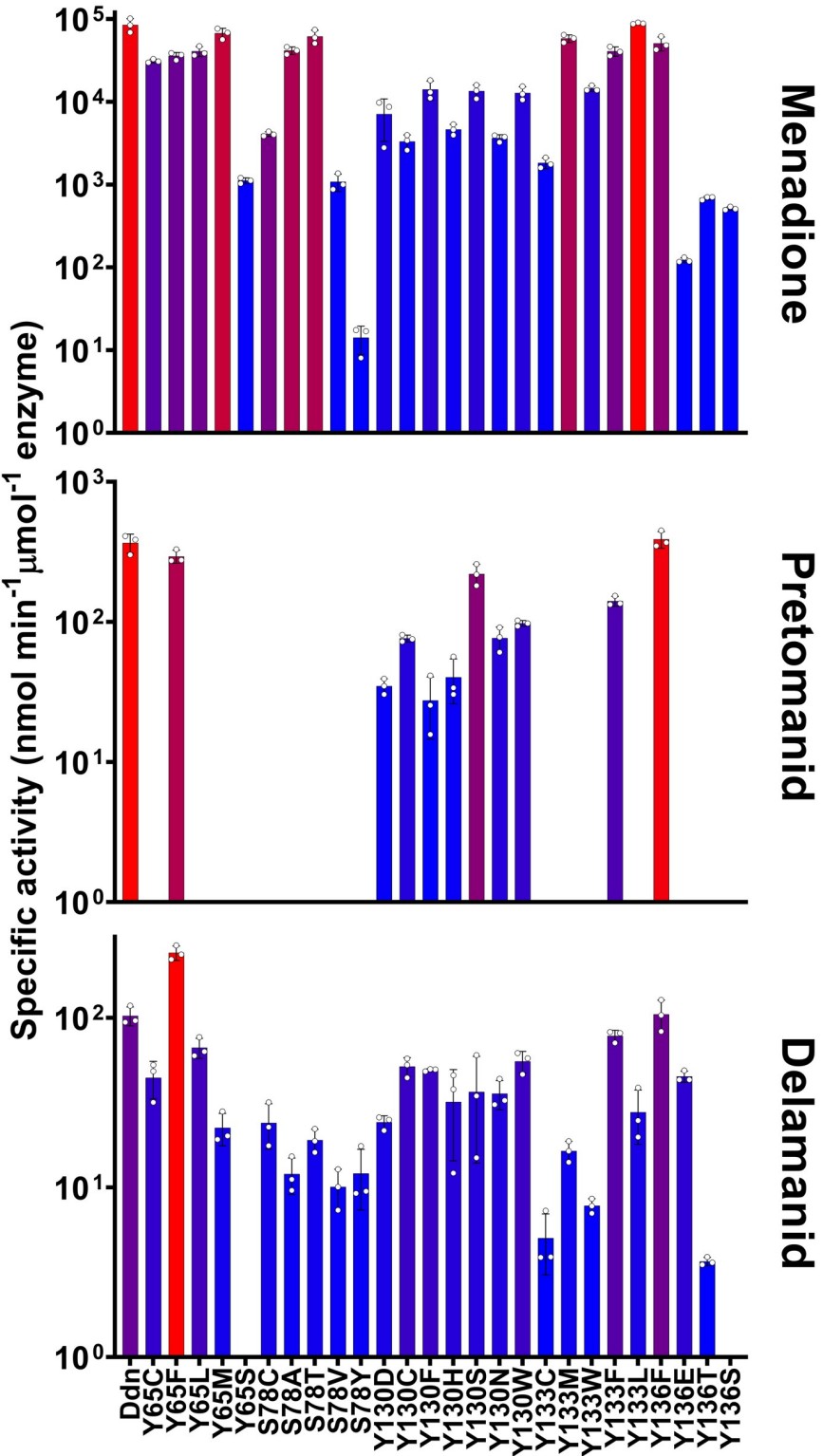

**Fig 6. Kinetic activities of mutants with potential spontaneous mutations within the substrate binding site of Ddn.** The activity of Ddn mutants with menadione, pretomanid, and delamanid. Bar color represents level of activity related to the rest of the graph with the highest activity colored red and the lowest activity colored blue. Error bars show standard deviations from three independent replicates. Data from this Figure is also present in Tables 2 and 3.

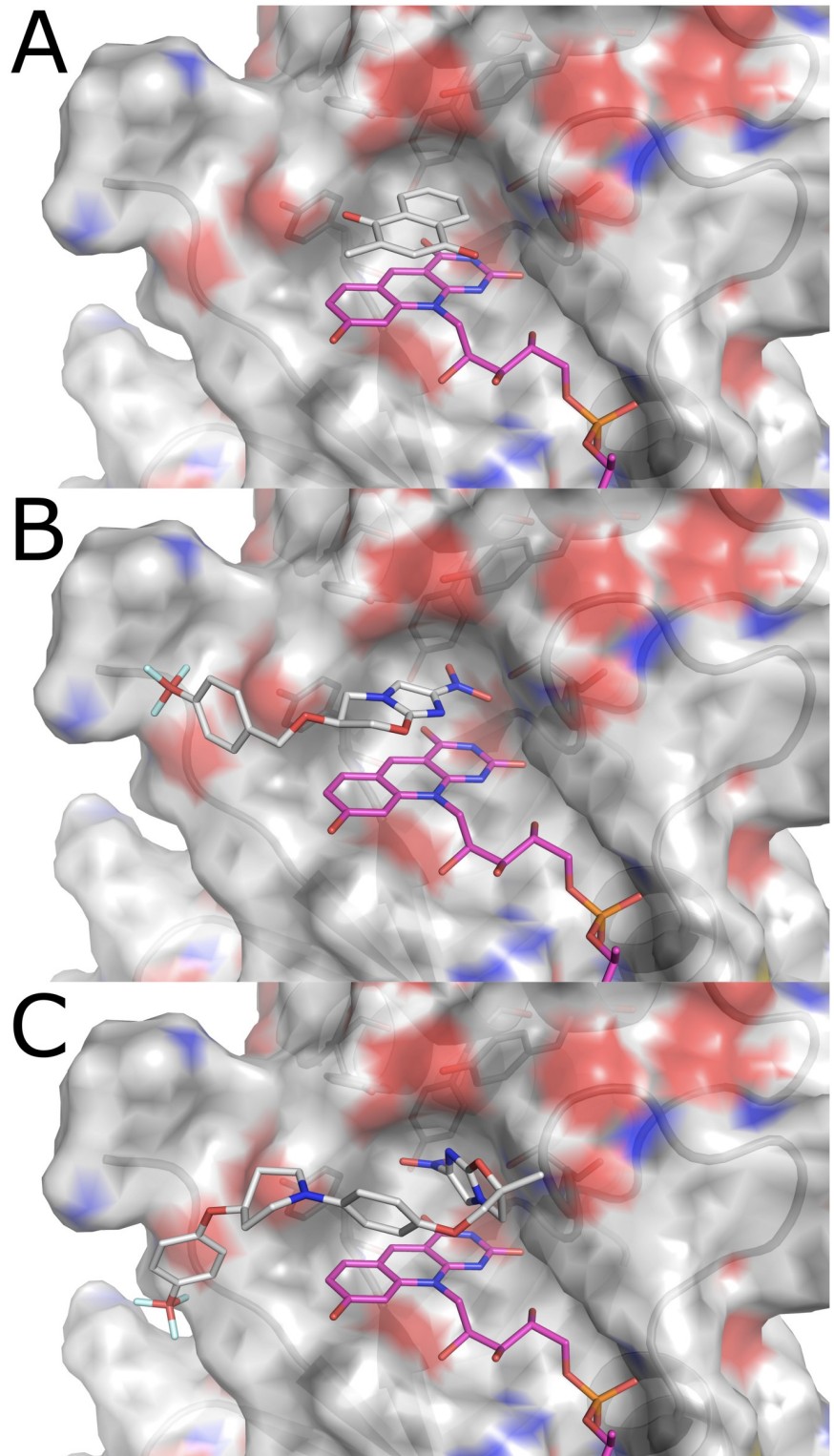

**Fig 7. The substrate-binding pocket of Ddn (PDB: 3R5R).** **(a)** Menadione (docking score = -6.584 kJ/mol) binds above the deazaflavin ring of $F_{420}H_2$ in a complementary pocket formed by the aromatic rings of the tyrosine residues and the hydroxyl group of Ser78. **(b)** Pretomanid (docking score = -5.995 kJ/mol) docked into wild-type Ddn with $F_{420}H_2$ bound. The nitroimidazo-oxazine moiety can bind parallel to the deazaflavin group of $F_{420}$, extending the nitro-group deep into the active site, towards Ser78. **(c)** Delamanid (docking score = -4.890 kJ/mol) docked into wild-

type Ddn with $F_{420}H_2$ bound. The dual methyl/phenoxy-methyl substitution at the C6 position of the oxazole ring creates steric hindrance with the deazaflavin ring of $F_{420}$, causing it to bind above $F_{420}$ in a perpendicular orientation. This results in additional distance to Ser78.

delamanid binding above the deazaflavin ring of $F_{420}$ in a ring-stacked orientation, as is seen in pretomanid, which has an oxazine ring with a single substituent in the analogous position. This results in a change in the angle at which the nitroimidazole group interacts with the cofactor that results in an increase in the distance to S78Y. Thus, there appears to be a plausible molecular basis for the differing effects of the S78Y mutation on pretomanid and delamanid activity *in vitro* and *in vivo*.

## Ddn mutants are virulent in mice but show defective recovery from hypoxic stress *in vitro*

To understand the fitness costs of loss of Ddn activity through mutation, we investigated the *in vitro* enzymatic properties of Ddn variants that have been identified in *M. tuberculosis*. Several isogenic pretomanid-resistant *M. tuberculosis* mutants have recently been selected by pretomanid monotherapy in infected mice and shown to harbour mutations in Ddn (M1T, L49P, L64P, R112W, C149Y and from insertion of the IS6110 transposable element at D108) [34]. We also investigated two mutants from a previous study in which resistant strains were identified from an *in vitro* selection experiment (S22L, W88R). We tested the ability of these eight mutants to reduce native (menadione) and drug (pretomanid and delamanid) substrates *in vitro*, finding that all mutations resulted in loss of detectable pretomanid activation, consistent with their selection in resistant strains. We observed that the M1T mutant (loss of start codon) and the mutants harbouring the IS6110 insertions did not produce any functional protein and therefore had no detectable native, nor prodrug-activating activity. In contrast, many of the point mutants, such as the L64P mutant, retained a small amount of activity with menadione, suggesting that it might retain a fraction of its native function (Table 2). Because many of these mutants were found in *M. tuberculosis* living in mouse lungs, the fitness cost of these mutations is clearly low enough to allow them to survive in that environment.

To understand the fitness costs of loss of Ddn activity in more detail, we investigated the ability of *M. tuberculosis* strains with mutations in Ddn to survive in different conditions. We therefore tested a selection of variants: wild-type H37Rv (full native activity), L64P (partial loss of native activity), as well as M1T and IS6110 (complete loss of native activity) to examine whether these mutants were attenuated for multiplication and survival in mice. These mutants showed no difference in growth or survival compared to wild type after low-dose aerosol infection of mice (Fig 8a), suggesting that these Ddn mutants would be transmissible.

A previous study showed that *M. tuberculosis* mutants that were unable to biosynthesise $F_{420}$ have a survival defect when recovering from hypoxia [12]. To better understand the competitive cost that loss of Ddn function causes to *M. tuberculosis* during recovery from hypoxia, we created a hygromycin marked deletion mutant in the mc$^2$6230 strain of *Mycobacterium tuberculosis* using phage transduction. The suspected Δ*ddn* mutants with hygromycin resistance displayed MIC values against delamanid of greater than 512 μM. The deletion mutants were then confirmed using whole genome sequencing (S1 Fig). The Δ*ddn* mutant displayed no growth or survival defects compared to the isogenic wild type when grown in aerobic or hypoxic conditions (Fig 8b and 8c) consistent with the previous results on *ddn* point mutants (Fig 8a). We then tested the marked Δ*ddn* strain in competition with the wild type isogenic strain when reaerated from hypoxia. As a control we took the day 7 aerobic growth sample of each strain and grew them in competition. We found that the Δ*ddn* mutant had lost no

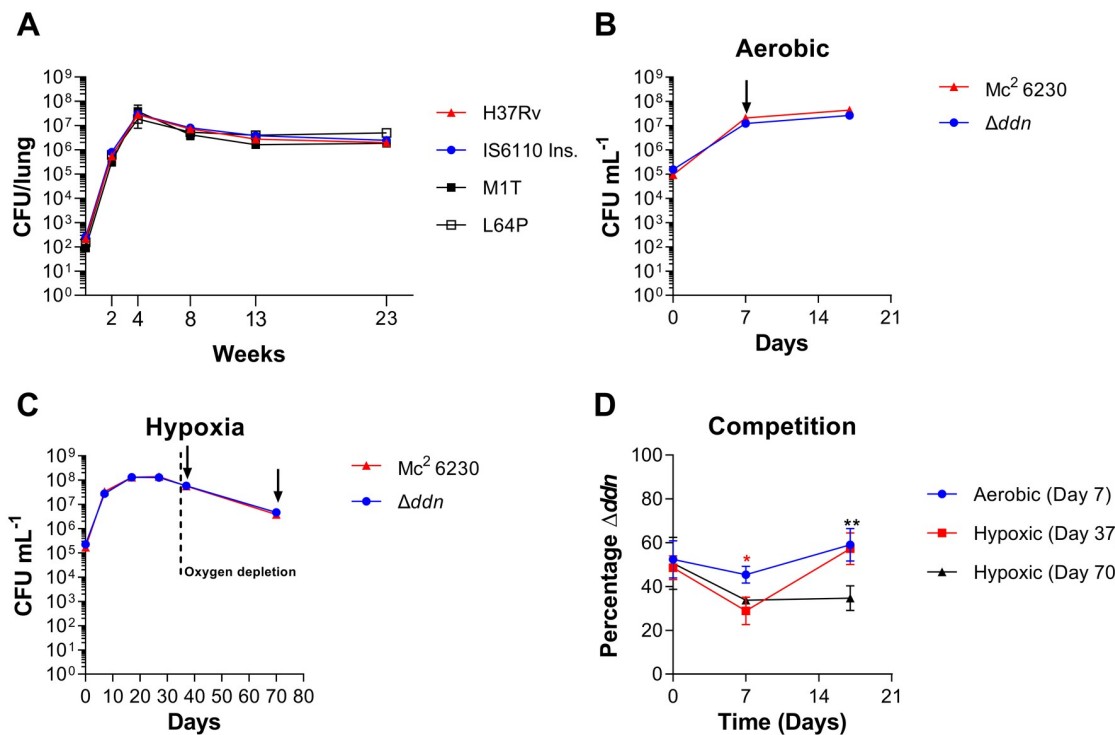

**Fig 8. Fitness analysis of Ddn mutants. (a)** *M. tuberculosis* Ddn mutants are not attenuated in mice. Mean lung colony-forming unit (CFU) counts of the H37Rv strain and isogenic Ddn mutants are shown following low-dose aerosol infection. Error bars represent SEM biological triplicate. Competition of the Δ*ddn* mutant compared to isogenic wild type mc² 6230. **(b)** Growth and survival of the Δ*ddn* mutant compared to isogenic wild type in aerobic conditions. **(c)** Growth and survival of the Δ*ddn* mutant compared to isogenic wild type in hypoxic conditions. (A and B) Arrows indicate samples taken for competition assay. **(d)** Percentage of the Δ*ddn* mutant when grown in competition with the isogenic wild type. Two-way ANOVA with Tukey multiple comparisons test where *, $p < 0.05$; **, $p < 0.01$. Red * is comparison between aerobic (Day 7) and Hypoxic (Day 37) at given timepoint, Black * is comparison between aerobic (Day 7) and Hypoxic (Day 70) at given timepoint. Error bars for b, c and d representative of standard deviation biological triplicate.

competitiveness when taken from aerobic conditions with the proportional percentage of Δ*ddn* remaining at about ~50% (Fig 8d). We then took equal proportions of each strain from hypoxia at 37 days and 70 days, which we grew in competition (Fig 8c). The proportion of Δ*ddn* was then measured at days 0, 7 and 17. We found that with the Day 37 hypoxic sample the Δ*ddn* mutant had an initial drop in proportion to 28% at 7 days but by day 17 the proportion of the Δ*ddn* mutant had returned to 57% (Fig 8d). The day 70 hypoxic sample had an initial drop off Δ*ddn* proportion to 33% which was maintained through until day 17 (Fig 8d). These data demonstrate that the Δ*ddn* mutant cannot compete with wild type after exit from prolonged hypoxia and that Ddn contributes to fitness under these conditions, perhaps because they are more susceptible to oxidative stress or have some defect in respiration.

## Discussion

### The fitness trade-off between resistance and native function

Given the conservation of Ddn-like genes throughout mycobacteria, its activity with menaquinone, and the inability of Ddn knock-outs to recover from hypoxia, we suggest that mutations that completely knock-out Ddn activity (including the loss of $F_{420}$ biosynthesis through mutations to $F_{420}$ biosynthetic genes, loss of $F_{420}$ reductase activity through knockout of FGD, and introduction of stop codons or large genetic insertions/deletions in *ddn*) could result in

substantial loss of fitness given that the ability to recover from hypoxia is an important aspect of *M. tuberculosis* pathogenesis [3,23,40–42]. Thus, for nitroimidazole resistance to spread and endanger health, the activity of Ddn must be either retained or otherwise compensated for. This means that mutations that are neutral or near-neutral for the wild-type activity, but knock out nitroimidazole-activation, will be of great clinical concern.

The binding of the prodrugs pretomanid and delamanid by Ddn, which leads to the activation of these prodrugs, is a promiscuous activity that is not fully coupled to its ability to bind and reduce its native substrate; i.e. loss of pretomanid activation is possible without total loss of the native activity. It has long been known that promiscuous activities of enzymes are less robust to the effects of mutations than native enzyme functions [39], meaning that it is possible that mutations could cause loss of prodrug activation without significant loss of native function. In this study we tested this assumption for Ddn, mutating ~1/3 of the amino acid positions in *M. tuberculosis* Ddn, including all known naturally occurring polymorphisms, showing that many such mutations (including several already present in clinical isolates) can cause loss of delamanid and/or pretomanid activation without loss of the native activity. Extending our enzymatic measurements to measurement of MICs revealed that related species with sequence differences in *ddn*, and clinical isolates of *M. tuberculosis* that harbor mutations within the active site, are resistant to pretomanid. These results have important implications for the clinical use of nitroimidazoles in TB treatment in order to optimize treatment outcomes and to prevent or slow the development of resistance. For instance, pretomanid will not be effective in patients infected with naturally occurring *M. tuberculosis* variants harbouring an S78Y mutation and indiscriminant use of pretomanid against such variants (such as in regions in which the N0008 strain is endemic) could drive selective amplification and spread of pretomanid resistance. Moreover, the chances of spontaneous mutation of Ddn in currently sensitive strains is significant, given our observation of the number of single nucleotide mutations that can knock out nitroreductase activity. Given that the clinical isolate N0008 is highly transmissible, it is likely that other Ddn variants in which nitroreductase activity can be abolished without substantial loss of the native activity will also be infectious owing to the minimal loss of fitness these mutations cause with the native activity. Thus, fitness-neutral Ddn mutations are likely to be a prominent route through which clinical pretomanid and delamanid resistance spreads.

Our findings have broad implications for the continued clinical development and usage of nitroimidazole antitubercular agents. Through multiple ongoing phase II/III clinical trials, the TB Alliance and others are evaluating novel combination therapies including pretomanid. The ongoing ZeNix and TB-PRACTECAL trials are extending the study of the BPaL (bedaquiline, pretomanid, linezolid) regimen that showed highly promising results against MDR- and XDR-TB in the Nix-TB trial, while the SimpliciTB trial studies the BPaMZ regimen (bedaquiline, pretomanid, moxifloxacin, pyrazinamide) against MDR- and drug-susceptible TB. Meanwhile, comparable delamanid-containing regimens are being studied in the endTB and MDR-END trials. Now that both delamanid and pretomanid are approved for clinical use, intensive monitoring of *ddn* polymorphisms is recommended to ensure informed regimen selection and allow interventions that will reduce spreading of transmissible resistance. Our findings indicate that delamanid binds to Ddn in a different conformation than pretomanid does, suggesting that it could be effective against some pretomanid-resistant isolates. It is also possible that combination therapy with both nitroimidazoles could help prevent the evolution and spread of resistance (since the simultaneous loss of both activities will be less likely to result from single amino acid substitution). Further studies on how delamanid and pretomanid are activated in the *M. tuberculosis* cell will inform the development of improved

nitroimidazole therapies and testing of a broad range of nitroimidazole analogs against a panel of Ddn variants could help to identify compounds for which resistance is less likely to evolve.

# Materials and methods

## Plasmid construction and point mutation

The *E. coli* codon optimised sequences for Ddn from *M. tuberculosis*, MSMEG_5998 from *M. smegmatis*, MMAR_5035 from *M. marinum*, MVAN_5261 from *M. vanbaalenii*, MAV_0613 from *M. avium*, and MUL_4109 from *M. ulcerans* were purchased as gene strings from ThermoFisher Scientific (Massachusetts, USA) and cloned into the expression vector pMAL-c2X using Gibson assembly [43]. All mutations to Ddn were made by site-directed mutagenesis using Gibson assembly [43]. Construction of MSMEG_2027 and FGD has been described previously [44,45].

## Protein expression and purification

MSMEG_2027 was expressed and purified as previously described [13]. Ddn, Ddn mutants, and Ddn orthologs were transformed into *E. coli* BL21 (DE3) cells and grown on LB agar containing 100 µg/ml ampicillin. Single colonies were picked and inoculated in LB media with 100 µg/ml ampicillin. Starter cultures were grown overnight and diluted 1/100 and grown at 37 ˚C until OD = 0.4. Cultures were induced with IPTG to a final concentration of 0.3 mM, and grown for 3 h at 25 ˚C. Cells were harvested by centrifugation at $8,500 \times g$ for 20 minutes at 4 ˚C and resuspended in lysis buffer (20 mM Tris-Cl, pH 7.5, 200 mM NaCl) and lysed by sonication using an Omni Sonicator Ruptor 400 (2 x 6 min. at 50% power). The soluble extract was obtained by centrifugation at $13,500 \times g$ for 1 h at 4 ˚C. The protein was purified using amylose resin (NEB) using the provided protocol. Briefly the lysate was passed over the amylose resin and washed with 12 column volumes of lysis buffer. The protein was eluted using elution buffer (same as lysis buffer but with 10 mM maltose). Samples were frozen at -80 ˚C in 20 mM tris pH 7.5, 200 mM NaCl, 10 mM maltose, and 10% glycerol.

FGD was expressed and purified as described by Bashiri *et al*. [45], with minor modifications. FGD was transformed into *E. coli* BL21 (DE3) cells and grown on LB agar containing 100 µg/ml ampicillin. Single colonies were picked and inoculated in Terrific Broth (TB) [46] with 100 µg/ml. ampicillin. Starter cultures were grown overnight and diluted 1/100 into auto-induction media (20 g/l tryptone, 5 g/l yeast extract, 5 g/l NaCl, 6 g/l $Na_2HPO_4$, 3 g/l $KH_2PO_4$, 6 ml/l glycerol, 2 g/l lactose, 0.5 g/l glucose, 100 mg/ml ampicillin) and grown at 30 ˚C for 24 h. The cells were harvested by centrifugation at 8500 g for 20 min at 4 ˚C and resuspended in lysis buffer (20 mM $NaPO_4$ pH 8, 300 mM NaCl, 25 mM imidazole) and lysed by sonication using an Omni Sonicator Ruptor 400 (2 x 6 min at 50% power). The soluble extract was obtained by centrifugation at 13500 g for 1 h at 4 ˚C. The soluble fraction was filtered and loaded 5-ml HisTrap HP column (GE Healthcare) and washed with lysis buffer. The protein was eluted with elution buffer (lysis buffer with 250 mM imidazole). The purified protein was dialyzed in 50 mM Tris-Cl, pH 7, 200 mM ammonium sulphate. Samples were frozen at -80 ˚C in 50 mM Tris-Cl, pH 7, 200 mM ammonium sulphate, and 10% glycerol.

## Enzymatic assays

$F_{420}$ was purified from M. *smegmatis* mc$^2$4517 as described by Ahmed *et al*.[13]. $F_{420}$ was reduced overnight with 10 µM FGD and 10 mM glucose-6-phosphate in 20 mM Tris-CL, pH 7.5 under anaerobic conditions. FGD was removed by spin filtration in a 1 mL 10 K MWCO spin filter (Millipore) and used $F_{420}H_2$ within 8 hours. Enzyme assays were performed

according to Ahmed *et al.* [13,47] in 200 mM Tris-HCl, pH 7.5, 0.1% Triton X-100, 25 μM $F_{420}H_2$, and 25 μM of substrate at room temperature. Enzyme concentrations used were between 0.1 μM and 1 μM. Activity was monitored following the oxidation of $F_{420}H_2$ which was measured spectrophotometrically at 420 nm ($\varepsilon = 41,400$ $M^{-1}$ $cm^{-1}$) [48]. Specific enzymatic activity with delamanid was measured using fluorescence (excitation/emission: 400 nm/ 470 nm).

## Computational analysis

Sequences of Ddn and orthologs were obtained from the NCBI sequence database. Alignment of the sequences were performed using MUSCLE [49] via the EMBL-EBI web services [50]. Autodock Vina [51] was used to dock menadione, pretomanid and delamanid into Ddn (PDB ID: 3R5R) [7]). The protein and ligand were prepared using in Autodock tools with default settings [52] and visualized using Pymol [53]. Substrate structures were obtained from the ZINC database [54].

## Drug susceptibility testing

Minimum inhibitory concentration (MIC) testing of *M. marinum* was performed according to the method of Wiegand *et al.* [55] with some minor modifications. A culture was grown on Brown and Buckle media and colonies were scraped and diluted to an $OD_{600}$ of 0.2, and was used as the inoculum for the MIC assay. Plates were incubated at 30 ˚C in a humid environment and were read after 5 days of incubation. MICs of *M. tuberculosis* isolates were determined as described previously [56] with slight modifications. Briefly, U-bottomed 96-well microtitre plates containing Middlebrook 7H9 broth supplemented with oleic acid, albumin, dextrose and catalase were inoculated with *M. tuberculosis* H37Rv or N0008 ($OD_{600}$ 0.02) in the presence of delamanid and pretomanid (0.015–512 μg/ml), and incubated at 37 ˚C for 5 days. The presence/absence of a cell pellet was checked visually. Resazurin (0.03% w/v) was then added, and plates were further incubated for 7 days. Viable cells reduce resazurin (blue) to resorufin (pink). Bedaquiline (0.015–32 μg/ml) was used as a drug susceptibility control.

## Dataset

To determine the frequency of non-synonymous sequence polymorphisms in the *ddn* gene we assembled a collection of 5,184 publicly available complete and draft *M. tuberculosis* genomes from Genbank [57] on the 7[th] of April 2017. Additionally, using the structured query https://www.ncbi.nlm.nih.gov/sra/(Illumina[Platform]) AND "Mycobacterium tuberculosis"[orgn:__txid1773]) to search the SRA (https://www.ncbi.nlm.nih.gov/sra/) opened on the 28[th] of April 2017, we identified 9,692 unassembled *M. tuberculosis* genome datasets.

## Lineage-typing of *M. tuberculosis* genomes

Classification of *M. tuberculosis* phylogenetic lineages was performed using KvarQ version 0.12.3a1 and default parameters [58]. Using KvarQ scan, unassembled sequencing data were screened against a database of previously defined sequence polymorphisms known to delineate the 7 major *M. tuberculosis* phylogenetic lineages (S1 Table).

## Characterisation of *ddn* sequence polymorphisms in *M. tuberculosis*

Using BLASTn (with default parameters) complete and draft *M. tuberculosis* genome assemblies from Genbank were queried with the *ddn* gene from *M. tuberculosis* H37Rv (Genbank accession:AL123456; [59]), identifying 145 *M. tuberculosis* genomes carrying alternative *ddn*

alleles (relative to H37Rv). For these 145 genomes the Ddn sequence was extracted and aligned using Clustal Omega version 1.2.4 [60] to identify non-synonymous mutation and other sequence polymorphism which could impact the function of Ddn (S1 Table).

For the unassembled genomes datasets, raw sequencing reads were aligned to the reference genome *M. tuberculosis* H37Rv using Snippy version 2.9 (BWA-mem version 0.7.12 [61]) (https://github.com/tseeman/snippy). Variant calling was performed using Snippy (Freebayes version 0.9.21 [62] with default parameters (minimum read coverage of 10x and 90% read concordance at the variant locus). 244 unassembled genome datasets were identified as having a mutation in *ddn*, only three of which were represented amongst the afore-mentioned 145 assembled genomes available in GenBank.

## Taxonomic profiling of unassembled sequencing data

Kraken [63], an ultrafast sequence classification tool, was used to assign taxonomic labels to the unassembled *M. tuberculosis* genomes from the SRA. Sequence reads for each genome from the SRA were screened against a NCBI refseq database (https://www.ncbi.nlm.nih.gov/refseq) using Kraken version 0.10.5 to identify contaminants. Of the 244 *M. tuberculosis* genomes with sequence polymorphisms in *ddn*, 67 (27.5%) were found to be contaminated with DNA from unidentified organisms or from *Mycobacterium* species other than *M. tuberculosis* (S2 Table). These genomes were excluded from all further analysis.

## Phylogenetic reconstruction of *M. tuberculosis* strains with sequencing polymorphisms in Ddn

To determine the phylogenetic distribution of *ddn* alleles in *M. tuberculosis* we carried out a phylogenomic analysis of *M. tuberculosis* strains with sequence polymorphism in *ddn*. Firstly, unassembled genomes from the SRA were assembled *de novo* using SPAdes version 3.9.0 [64]. Next, all 322 complete and draft genomes (S3 Table) were aligned with Parsnp version 1.2 [65] using the genome of H37Rv as a reference to produce a core genome alignment of 848,476 bp. Core genome SNPs were identified and recombinant regions removed using Gubbins version 2.2.0 [66]. Finally, a maximum-likelihood phylogenetic tree was estimated using RAxML version 8.2.9 [67] under the GTRGAMMA nucleotide substitution model.

## Evaluation of *ddn* mutant fitness in low-dose aerosol mouse infection model

Isogenic pretomanid-resistant mutants were selected in the *M. tuberculosis* H37Rv strain background and subjected to whole genome sequencing, as previously described [34]. In the present study, female BALB/c mice were aerosol-infected with the H37Rv parent strain or one of the following isogenic *ddn* mutants, implanting approximately 2 $\log_{10}$ CFU/lung in each case: L64P, M1T, or IS6110 insertion at D108. Three mice from each group were sacrificed on the day after infection to confirm the number of CFU implanted. The multiplication and persistence of each strain was evaluated by sacrificing 3 mice/group and quantifying the CFU counts by plating serial dilutions of lung homogenate on Middlebrook selective 7H11 agar at 2, 4, 8, 13 and 23 weeks post-infection.

## Evaluation of *ddn* mutant fitness during and after hypoxic stress

The H37Rv parent and isogenic *ddn* mutants used for mouse infection were inoculated into tubes containing Dubos Tween Albumin Broth (BD Difco) supplemented with the hypoxia indicator dye, methylene blue (500 μg/ml). Tubes were sealed with rubber stoppers and

incubated at 37 ˚C with slow magnetic stirring. The reduction and decolorization of methylene blue served as a visual indicator of oxygen levels corresponding to NRP stage 2 [68,69], which was attained in all cultures by day 24. Aliquots were removed for quantitative culture on 7H11 agar after 15, 24, 29, 43 and 57 days of incubation. Samples were collected by piercing the rubber stoppers with a syringe and sampling did not introduce atmospheric oxygen into the tubes, as shown by maintenance of the decolorized dye state. After day 57, each strain was returned to normoxic conditions by transferring the contents of the tubes to sterile Erlenmeyer flasks and incubated at 37 ˚C in a shaker for another 14 days [70]. Aliquots were removed for quantitative culture after 7 and 14 days of normoxia.

## DDN marked deletion construction

To generate the DDN Hygromycin marked deletion mutant we used a high throughput phage-based method based on Jain et al (2014) [71]. The phasmid construct for this deletion was a gift from William R. Jacobs. The DDN deletion phasmid DNA was electroporated into electrocompetent *M. smegmatis*. The *M. smegmatis* was recovered for two hours in 7H9, then split and mixed into two 7H9 agar overlay mixtures (7H9 powder + 0.3% agar) and spread on 7H11 plates. The plates were incubated for two to three days at 30˚C to allow phage growth and lysis. Resulting plaques were picked using a pasteur pipette and resuspended in MP buffer (50 mM Tris, 150 mM NaCl, 10 mM MgSO4, 2 mM CaCl2, pH 7.6–7.8). The MP buffer containing the phage was split and mixed with two *M. smegmatis* cultures, followed by mixing into two 7H9 agar overlay mixtures. The overlay mixes were spread on 7H11 plates and examined for plaques after two to three days at 30˚C. This resulted in an increased number of plaques, which were harvested by pipetting MP buffer on the of the plate and shaking at 100 rpm for 3 hours. The MP buffer containing the phage was harvested from the plate with a syringe before being passed through a 20 μm filter to remove the bacteria. The resulting sterile MP buffer with phage titre was quantified by serial dilution followed by spot plating onto dry *M. smegmatis* overlay plates. The phage amplification was repeated until a titre of $10^9$, $10^{10}$ phage was achieved. mc$^2$ 6230 *Mycobacterium tuberculosis* was grown up in ink wells to an optical density (600 nm) of 0.6 to 0.8, before harvest at 3,126 x g. Cells were then transduced by resuspension in the MP buffer containing the phage titre at 37 ˚C (to prevent phage lysing the cell) overnight followed by plating on 7H11 with 50 μg/mL hygromycin. Colonies with hygromycin resistance were indicative of successful double homologous recombination with the gene being replaced with hygR-*sacB* DNA fragment. The Δ*ddn* mutant was screened for resistance to delamanid. The Δ*ddn* mutant was confirmed through whole genome sequencing (Illumina 150 bp paired end reads) with reads mapped using the Bowtie2 plugin for Geneious to the wild type H37RV genome (S1a Fig) and a H37RV genome with the expected hygromycin marker (S1b Fig).

## Competition assays

For the competition assay aerobically grown cultures the Δ*ddn* mutant and wild type mc$^2$ 6230 were grown in 7H9 in separate inkwells and monitored by spot platting 5 μL of a serial dilution in triplicate on 7H11 agar plates. At 7 days a new combined inkwell was inoculated with 100μL of each strain. For the competition assay from hypoxically grown cultures the Δ*ddn* mutant and wild type mc$^2$ 6230 were grown in separate 30 mL serum vials in 7H9 and monitored by spot platting 5 μL of a serial dilution in triplicate on 7H11 agar plates. Oxygen depletion was estimated by de-colourisation of 1.5 μg/mL methylene blue which occurred between 17–27 days. At 37 days 100 μL of the Δ*ddn* mutant and wild type mc$^2$ 6230 was taken and used to inoculate a single inkwell with 10 mL 7H9. At 70 days 200 μL of the Δ*ddn* mutant and wild type mc$^2$ 6230 was taken and used to inoculate a single inkwell with 10 mL 7H9. The combined

inkwells with both strains were assayed for the proportion of the Δ*ddn* mutant by spot platting 5 μL of a serial dilution in triplicate on 7H11 agar plates with and without 50 μg/mL hygromycin. The Δ*ddn* mutant proportion of total cells was established from the number of colonies on the hygromycin agar plates and expressed as a percentage of total colonies on the plates with no hygromycin.

## Supporting information

**S1 Fig.** (A) Map of the Δ*ddn* mutant reads using Bowtie 2 to H37Rv genome. (B) Map of the Δ*ddn* mutant reads using Bowtie 2 to H37Rv genome with expected hygromycin marker. (TIF)

**S1 Table. Distribution of Ddn alleles in the major TB lineages.** (XLSX)

**S2 Table. Taxonomic classification of sequence reads from bacterial genome data sets described as M. tuberculosis on the sequence read archive (SRA) using Kraken: Red shading indicates genome datasets contaminated with DNA from unknown organisms (Reads classified (%)) or datasets where the majority of reads, identified as genus Mycobacterium, match to a Mycobacterium species other than M. tuberculosis.** (XLSX)

**S3 Table. *M. tuberculosis* genome datasets used in this study.** (XLSX)

## Acknowledgments

We acknowledge Thomas Cuddihy (QFAB Bioinformatics) for assistance with genome data retrieval. We thank William R. Jacobs Jr. for the gift of the phasmid used to make the Δ*ddn* mutant. We thank Sebastien Gagneux for the gift of the *M. tuberculosis* N0008 strain.

## Author Contributions

**Conceptualization:** Brendon M. Lee, Livnat Afriat-Jurnou, Colin J. Jackson.

**Data curation:** Brendon M. Lee, Liam K. Harold, Deepak V. Almeida, Htin Lin Aung, Brian M. Forde, Sacha J. Pidot, Timothy P. Stinear, Scott A. Beatson, Eric L. Nuermberger, Gregory M. Cook, Colin J. Jackson.

**Formal analysis:** Brendon M. Lee, Liam K. Harold, Deepak V. Almeida, Brian M. Forde, Kiel Hards, Sacha J. Pidot, Timothy P. Stinear, Chris Greening, Scott A. Beatson, Eric L. Nuermberger, Gregory M. Cook, Colin J. Jackson.

**Funding acquisition:** Deepak V. Almeida, Chris Greening, Scott A. Beatson, Eric L. Nuermberger, Gregory M. Cook, Colin J. Jackson.

**Investigation:** Brendon M. Lee, Liam K. Harold, Deepak V. Almeida, Livnat Afriat-Jurnou, Htin Lin Aung, Brian M. Forde, Kiel Hards, Sacha J. Pidot, F. Hafna Ahmed, A. Elaaf Mohamed.

**Methodology:** Brendon M. Lee, Liam K. Harold, Deepak V. Almeida, Livnat Afriat-Jurnou, Htin Lin Aung, Brian M. Forde, Kiel Hards, Sacha J. Pidot, F. Hafna Ahmed, A. Elaaf Mohamed, Timothy P. Stinear, Scott A. Beatson, Eric L. Nuermberger, Gregory M. Cook, Colin J. Jackson.

**Project administration:** Brendon M. Lee, Scott A. Beatson, Eric L. Nuermberger, Gregory M. Cook, Colin J. Jackson.

**Resources:** Timothy P. Stinear, Scott A. Beatson, Eric L. Nuermberger, Gregory M. Cook, Colin J. Jackson.

**Supervision:** Kiel Hards, F. Hafna Ahmed, Matthew C. Taylor, Nicholas P. West, Timothy P. Stinear, Chris Greening, Scott A. Beatson, Eric L. Nuermberger, Gregory M. Cook, Colin J. Jackson.

**Visualization:** Brendon M. Lee, Liam K. Harold, Deepak V. Almeida, Brian M. Forde, Kiel Hards, Chris Greening, Scott A. Beatson, Eric L. Nuermberger, Gregory M. Cook, Colin J. Jackson.

**Writing – original draft:** Brendon M. Lee, Colin J. Jackson.

**Writing – review & editing:** Brendon M. Lee, Livnat Afriat-Jurnou, Brian M. Forde, Kiel Hards, F. Hafna Ahmed, Matthew C. Taylor, Nicholas P. West, Timothy P. Stinear, Chris Greening, Scott A. Beatson, Eric L. Nuermberger, Gregory M. Cook, Colin J. Jackson.

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
