## [Decision Letter · Decision Letter 0]

6 Aug 2019

Dear Mr Lee,

Thank you very much for submitting your manuscript "Predicting nitroimidazole antibiotic resistance mutations in Mycobacterium tuberculosis with protein engineering" (PPATHOGENS-D-19-01241) for review by PLOS Pathogens. Your manuscript was fully evaluated at the editorial level and by independent peer reviewers. The reviewers appreciated the attention to an important problem, but raised some substantial concerns about the manuscript as it currently stands. These issues must be addressed before we would be willing to consider a revised version of your study. We cannot, of course, promise publication at that time.

We therefore ask you to modify the manuscript according to the review recommendations before we can consider your manuscript for acceptance. Your revisions should address the specific points made by each reviewer.

(1) A letter containing a detailed list of your responses to the review comments and a description of the changes you have made in the manuscript. Please note while forming your response, if your article is accepted, you may have the opportunity to make the peer review history publicly available. The record will include editor decision letters (with reviews) and your responses to reviewer comments. If eligible, we will contact you to opt in or out.

(2) Two versions of the manuscript: one with either highlights or tracked changes denoting where the text has been changed; the other a clean version (uploaded as the manuscript file).

Additionally, to enhance the reproducibility of your results, PLOS recommends that you deposit your laboratory protocols in protocols.io, where a protocol can be assigned its own identifier (DOI) such that it can be cited independently in the future. For instructions see http://journals.plos.org/plospathogens/s/submission-guidelines#loc-materials-and-methods

We hope to receive your revised manuscript within 60 days. If you anticipate any delay in its return, we ask that you let us know the expected resubmission date by replying to this email. Revised manuscripts received beyond 60 days may require evaluation and peer review similar to that applied to newly submitted manuscripts.

[LINK]

Sincerely,

Helena Ingrid Boshoff

Associate Editor

PLOS Pathogens

JoAnne Flynn

Section Editor

PLOS Pathogens

Kasturi Haldar

Editor-in-Chief

PLOS Pathogens

orcid.org/0000-0001-5065-158X

Grant McFadden

Editor-in-Chief

PLOS Pathogens

orcid.org/0000-0002-2556-3526

The question whether this enzyme is a specific menaquinone reductase is relevant. Thus, it is warranted to probe a few substrates as suggested. In addition, it is important to note that fitness was never demonstrated. Measurement of fitness by a competitive assay under the relevant in vitro conditions could substantially strengthen this work. This can be done in vitro for the hypoxia work with the appending conclusions then limited to exit from hypoxia by performing fitness assessment (competition assay of fit versus impaired fitness mutants) with analysis done by allele-specific PCR. An ideal experiment would of course be competitive experiments in mice with for example 1:5 and 5:1 ratios to see which strain gets out-competed since this would really substantiate the claims about in vivo fitness.

Reviewer's Responses to Questions

**Part I - Summary**

Reviewer #1: This manuscript by Lee and colleagues presents a careful and fairly thorough molecular and biochemical analysis of the role of Ddn variants in resistance of Mycobacterium tuberculosis to pretomanid and delamanid. The authors make the interesting observation that reduction of menaquinone (menadione), pretomanid and delamanid have distinct structural requirements. The implication of this finding is that resistance to pretomanid and delamanid can occur without ablation of native function of Ddn. Therefore, resistance need not impair fitness. Moreover, while there is overlap in some requirements for pretomanid and delamanid reduction, some variant retain activity for one but not the other. Thus, resistance to one does not necessarily confer resistance to both. Importantly, the authors interrogate numerous genome sequences from clinical isolates, many of which are not expected to have been exposed to pretomanid and delamanid. The authors are able to confirm resistance based on their variant analysis. This study is important and interesting, and the study was well designed and well presented. I have some minor suggestions to improve clarity for readers.

Reviewer #2: This manuscript aims at predicting and understanding the role of resistance to nitroimidazole compounds by mutation in Mycobacteirum tuberculosis. One of the great strengths of this work is the use of natural polymorphisms in a large number of M. tuberculosis strains as a source of natural variability, which is of course highly relevant, given that nitroimidazole drugs will be used against such strains in practice. Most of the experiments appear to have been carefully designed, executed, analysed and interpreted.

Reviewer #3: Background: The rapid emergence of drug resistant variants of the causative agent of tuberculosis (TB), Mycobacterium tuberculosis, has created an urgent need for new therapeutic options for this disease. Among the new and repurposed drugs that have been introduced for TB are the nitroimidazoles, delamanid and pretomanid, which are prodrugs that require activation by the deazaflavin-dependent nitroreductase (Ddn). Resistance to delamanid is acquired through mutations in the biosynthesis of F420, which serves as a cofactor for Ddn, or by direct abrogation/reduction in Ddn activity. With nitroimidazoles and another new TB drug, bedaquiline, resistance has been reported clinically in individuals who have not been exposed to the drug. Considering this, the authors set out to determine the effect of possible resistance conferring mutations on Ddn activity and the resulting resistance to nitroimidazoles.

Summary of findings: The authors first demonstrate that Ddn displays F420-dependent menaquinone-1 reductase activity that appears to be somewhat weak in vitro and they speculate the activity within the native membrane environment in M. tuberculosis is most likely enhanced. In addition to M. tuberculosis, the authors demonstrate that Ddn orthologs from a variety of mycobacterial species display menaquinone-1 reductase activity but only Ddn from M. tuberculosis and M marinum can activate nitroimidazoles. The fact that Ddn reductase activity can be found without nitroimidazole activation activity suggests that the former function is not coupled to the latter. Consistent with this, those organisms from which Ddn was unable to activate nitroimidazoles were naturally resistant to these compounds. The authors then interrogated genomes of M. tuberculosis isolates and report 46 non-synonymous substitutions and 2 deletions in ddn. They engineer these mutations and create a panel of recombinant proteins. Using this, the authors demonstrate that these mutants retained menadione reductase activity and the majority of these were also able to activate pretomanid. Ddn mutations L49P, S78Y, K79Q, W88R, and Y133C were unable to activate pretomanid, two of these - S78Y and Y113C - were able to activate delaminid. The authors conduct a phylogenetic analysis of these mutations in clinical isolates and suggest that as S78Y mutation is present in clinically successful strains, it does not have a large fitness cost. In a Beijing isolate with the S78Y mutation, the authors demonstrate robust delamanid resistance. Next, the authors create a panel of recombinant Ddn mutants with point mutations in the substrate binding site and test these for native activity and nitroimidazole activation. From the 26 mutants, all retain varying levels of menadione activity and 16 did not display the ability to activate pretomanid. Using molecular docking, the authors demonstrate that binding of pretomanid to Ddn is different to delamanid. Using mutant strains of M. tuberculosis that were selected from mice on pretomanid monotherapy and from a previous study, the authors demonstrate that some of the point mutations identified earlier (M1T, L49P, L64P, R112W, C149Y, S22L, W88R) in their analysis are sufficient to confer resistance and were associated with reduced pretomanid activation. Some of these mutants were defective in their ability to recover from hypoxia with no associated survival defects under these conditions and in the murine model of TB infection.

**Part II – Major Issues: Key Experiments Required for Acceptance**

Reviewer #1: None

Reviewer #2: The authors need to improve their description of the enzymology results and its interpretations.

For example, on the first paragraph of results, the authors "Ddn catalyzed menaquinone reduction

in vitro with moderate efficiency (kcat/KM = 8.6 x 102 M-1 s-1)...". 10^2 M-1s-1 is not moderate efficiency, it is low efficiency. One would expect a value at around 10^6 M-1s-1 for a metabolic enzyme. Also, " and physiologically relevant affinity (KM = 22.4 ± 3.8 μM) (Table 1);", what is the concentration of menaquinone in the cells? Only by knowning that, the authors can judge if this is or isn't physiologic. Finally, Km does not equal to Kd, in most cases I have seen. Therefore, the authors must swap "affinity" by "Km".

On the second paragraph of the results, the authors leap to a conclusion without any supporting data, by stating "Having established that Ddn is a menaquinone reductase,...". To this point, the authors demonstrated that Ddn can reduce menaquinone, but that does not prove that Ddn is a menaquinone reductase. To prove that, the authors must test a number of other substrates for enzymes that reduce quinone and show that kcat/Km values are smaller than what they get for menaquinone. Showing that an enzyme act (very poorly) on a substrate does not prove anything. The authors must use a variety of other quinone reductase substrates to probe the substrate specificity and therefore assign the likely bona fide substrate specificity of Ddn, if that is what they are claiming they are doing.

All other parts of the manuscript appear to be well interpreted.

Reviewer #3: This is an exceptional biochemical analysis of amino acid sequence - structure - function relationships in Ddn with respect to its native reductase activity and the ability to activate nitroimidazoles. The study is well-executed from a biochemical perspective and has important implications for surveillance of the reported mutations in the clinical setting which would ultimately assist in better stewardship of nitroimidazoles as therapeutic agents for TB. That said, there are some shortfalls, detailed below

1. The authors use the term fitness several times but make no substantive measure of fitness in laboratory experiments. Most conclusions are derived from clinical and epidemiological information. Several aspects may be at play in the clinical settings and these remarks should be reconsidered. The claims of fitness costs or lack thereof are not support by the data presented in this manuscript.

2. Similarly, hypoxia seems to used interchangeably with dormancy. Dormancy is a complex phenomenon associated with metabolic quiescence. It is unclear if hypoxia in a test tube faithfully reflects this state. Moreover, recovery from such conditions in the lab is not a measure of fitness or transmissibility. This is all speculative and should be revised.

**Part III – Minor Issues: Editorial and Data Presentation Modifications**

Reviewer #1: 1. One of the interesting and important points of the manuscript is that resistance to these new bicyclic nitroimidazoles potentially occurs through genetic drift rather than by selection. The authors might consider bringing this point into the title and abstract.

2. In author summary, change "Strikingly, when analysed >15,000 M. tuberculosis..." to "Strikingly, when we analysed >15,000 M. tuberculosis..."

3. "In order to spread effectively and endanger health, these resistant strains would need to retain sufficient fitness to survive all stages of the lifecycle of M. tuberculosis, including recovery from dormancy." Formal proof of the importance of dormancy for fitness of M. tuberculosis in the host is non-existent. We know that such populations exist, but, whether this mode is essential for transmission and pathogenesis is unknown. The authors should modify this sentence.

4. Change "M. tuberculosis H37Rv and M. marinum, which were the only two strains that encoded Ddn orthologs with in vitro pretomanid activation activity," to "M. tuberculosis (H37Rv) and M. marinum, which were the only two species that encoded Ddn orthologs with in vitro pretomanid activation activity,"

5. "Every mutant tested was able to reduce menadione, indicating that there was selective pressure to maintain the physiological function of Ddn." Probably true, but, without the isoprenyl tail, menadione may be a more promiscuous substrate, it is overreaching to state that this observation indicates maintenance of function, would need to test MQ (but not necessary to do this). May be better to state that it is consistent with selective pressure for maintenance of the physiological function of Ddn...

6. "could readily occur and spread with sufficient selection pressure." to "could readily occur and spread with sufficient selective pressure."

7. "...will result in substantial loss of fitness given that the ability to recover from dormancy is such an important aspect of M. tuberculosis pathogenesis." While this might be the case, to my knowledge, there is no direct evidence, and the references provided do not provide such evidence. Maintenance of this function supports its role in fitness, but, that role might be distinct from that which has been modeled in the laboratory.

8. "The activation of the prodrugs pretomanid and delamanid by Ddn is a promiscuous activity that is not coupled to its native activity." Statement needs modification, hydride transfer is part of the native activity of Ddn

9. "It has long been known that promiscuous activities are more susceptible to mutation than native functions..." Modify, activities don't mutate.

10. Figure 3 title, "sequence" and "sequences" is redundant, for figure 3 legend, should describe what color represents.

11. Figure 4, increase font size of SNPs

12. Table 2 seems redundant with figures that show these data. However, I do favor inclusion -perhaps as supplemental. Also, the meaning of no activity (below limit of detection?) and no expression is unclear.

13. Table 3, rate of 0 should be expressed as less than the limit of detection. Need to define N/A, not sure why it says N/A for M. marinum when Table 2 shows activity.

14. Table 4 seems redundant with figure 4.

15. Same data is shown in multiple figures and tables, most journals require footnotes for repeated data.

16. For modeling, please show top docking scores.

17. Figure 7, replicates and statistics?

18. Need to provide additional information for readers to understand what is presented in Table S2.

19. Check th

Reviewer #2: There are a few typos in the manuscript. For example, on Figure 1A (M. ulcerans, not M. ulceran) and the title of the figure (binding, not bindng).

Please, change "Wt" by the strain name used, as for this case, clearly H37Rv or analogous strain is not a wild-type.

Reviewer #3: (No Response)

PLOS authors have the option to publish the peer review history of their article (what does this mean?). If published, this will include your full peer review and any attached files.

Reviewer #1: Yes: Anthony Baughn

Reviewer #2: Yes: Luiz Pedro Carvalho

Reviewer #3: No

---

## [Editor Report · Decision Letter 1]

16 Dec 2019

Dear Mr Lee,

We are pleased to inform that your manuscript, "Predicting nitroimidazole antibiotic resistance mutations in Mycobacterium tuberculosis with protein engineering", has been editorially accepted for publication at PLOS Pathogens. 

Before your manuscript can be formally accepted and sent to production, you will need to complete our formatting changes, which you will receive by email within a week. Please note that your manuscript will not be scheduled for publication until you have made the required changes.

IMPORTANT NOTES

(1) Please note, once your paper is accepted, an uncorrected proof of your manuscript will be published online ahead of the final version, unless you’ve already opted out via the online submission form. If, for any reason, you do not want an earlier version of your manuscript published online or are unsure if you have already indicated as such, please let the journal staff know immediately at plospathogens@plos.org.

(2) Copyediting and Proofreading: The corresponding author will receive a typeset proof for review, to ensure errors have not been introduced during production. Please review the PDF proof of your manuscript carefully, as this is the last chance to correct any errors. Please note that major changes, or those which affect the scientific understanding of the work, will likely cause delays to the publication date of your manuscript. 

(3) Appropriate Figure Files: Please remove all name and figure # text from your figure files. Please also take this time to check that your figures are of high resolution, which will improve the readbility of your figures and help expedite your manuscript's publication. Please note that figures must have been originally created at 300dpi or higher. Do not manually increase the resolution of your files. For instructions on how to properly obtain high quality images, please review our Figure Guidelines, with examples at: http://journals.plos.org/plospathogens/s/figures.

(4) Striking Image: Please upload a striking still image to accompany your article if one is available (you can include a new image or an existing one from within your manuscript). Should your paper be accepted, this image will be considered for our monthly issue image and may also appear on our website to feature your article. Please upload this as a separate file, selecting "striking image" as the file type upon upload. Please also include a separate "Other" file with a caption, including credits and any potential copyright information. Please do not include the caption in the main article file. If your image is from someone other than yourself, please ensure that the artist has read and agreed to the terms and conditions of the Creative Commons Attribution License at http://journals.plos.org/plospathogens/s/content-license. Please note that PLOS cannot publish copyrighted images.

(5) Press Release or Related Media: If your institution or institutions have a press office, please notify them about your upcoming paper at this point, to enable them to help maximize its impact. If they will be preparing press materials for this manuscript, please inform our press team in advance at plospathogens@plos.org as soon as possible. We ask that you contact us within one week to plan ahead of our fast Production schedule. If you need to know your paper's publication date for related media purposes, you must coordinate with our press team, and your manuscript will remain under a strict press embargo until the publication date and time. This means an early version of your manuscript will not be published ahead of your final version. 

(6)  PLOS requires an ORCID iD for all corresponding authors on papers submitted after December 6th, 2016. Please ensure that you have an ORCID iD and that it is validated in Editorial Manager.  To do this, go to ‘Update my Information’ (in the upper left-hand corner of the main menu), and click on the Fetch/Validate link next to the ORCID field.  This will take you to the ORCID site and allow you to create a new iD or authenticate a pre-existing iD in Editorial Manager

(7) Update your Profile Information: Now that your manuscript has been provisionally accepted, please log into Editorial Manager and update your profile, if needed. Go to https://www.editorialmanager.com/ppathogens, log in, and click on the "Update My Information" link at the top of the page. Please update your user information to ensure an efficient production and billing process. 

(8) LaTeX users only: Our staff will ask you to upload a TEX file in addition to the PDF before the paper can be sent to typesetting, so please carefully review our Latex Guidelines http://journals.plos.org/plospathogens/s/latex in the meantime.

(9) If you have associated protocols in protocols.io, please ensure that you make them public before publication to guarantee immediate access to the methodological details.

Best regards,

Helena Ingrid Boshoff

Associate Editor

PLOS Pathogens

JoAnne Flynn

Section Editor

PLOS Pathogens

Kasturi Haldar

Editor-in-Chief

PLOS Pathogens

orcid.org/0000-0001-5065-158X

Grant McFadden

Editor-in-Chief

PLOS Pathogens

orcid.org/0000-0002-2556-3526

The authors have addressed the reviewers' concerns. This is a manuscript that describes the importance of Ddn to pathogenesis based on its role in respiration. The results demonstrate which mutations confer resistance to Pretomanid or Delamanid without impairing the respiratory function of the enzyme.
---

## [Editor Report · Acceptance letter]

4 Feb 2020

Dear Mr Lee,

We are delighted to inform you that your manuscript, "Predicting nitroimidazole antibiotic resistance mutations in Mycobacterium tuberculosis with protein engineering," has been formally accepted for publication in PLOS Pathogens.

Best regards,

Kasturi Haldar

Editor-in-Chief

PLOS Pathogens

orcid.org/0000-0001-5065-158X

Michael Malim

Editor-in-Chief

PLOS Pathogens

orcid.org/0000-0002-7699-2064